# DP-SGD Without Clipping: The Lipschitz Neural Network Way

Louis Béthune[*†]    Thomas Masséna[*‡]    Thibaut Boissin[*‡]    Aurélien Bellet [§]

Franck Mamalet [‡]    Yannick Prudent [‡]    Corentin Friedrich [‡]    Mathieu Serrurier [†]

David Vigouroux [‡]

## ABSTRACT

State-of-the-art approaches for training Differentially Private (DP) Deep Neural Networks (DNN) face difficulties to estimate tight bounds on the sensitivity of the network's layers, and instead rely on a process of per-sample gradient clipping. This clipping process not only biases the direction of gradients but also proves costly both in memory consumption and in computation. To provide sensitivity bounds and bypass the drawbacks of the clipping process, we propose to rely on Lipschitz constrained networks. Our theoretical analysis reveals an unexplored link between the Lipschitz constant with respect to their input and the one with respect to their parameters. By bounding the Lipschitz constant of each layer with respect to its parameters, we prove that we can train these networks with privacy guarantees. Our analysis not only allows the computation of the aforementioned sensitivities at scale, but also provides guidance on how to maximize the gradient-to-noise ratio for fixed privacy guarantees. The code has been released as a Python package available at `https://github.com/Algue-Rythme/lip-dp`

## 1 INTRODUCTION

The field of differential privacy seeks to enable deep learning on sensitive data, while ensuring that models do not inadvertently memorize or reveal specific details about individual samples in their weights. This involves incorporating privacy-preserving mechanisms into the design of deep learning architectures and training algorithms, whose most popular example is Differentially Private Stochastic Gradient Descent (DP-SGD) (Abadi et al., 2016). One main drawback of classical DP-SGD methods is that they require costly per-sample backward processing and gradient clipping. In this paper, we offer a new method that unlocks fast differentially private training through the use of Lipschitz constrained neural networks.

**Differential privacy.** Informally, differential privacy (DP) is a *definition* that bounds how much the change of a single sample in a dataset affects the range of a stochastic function (here, the training algorithm). This definition is largely accepted as a strong guarantee against privacy leakages under various scenarii, including data aggregation or post-processing (Dwork et al., 2006).

In this paper we focus on supervised learning tasks: we assume that the dataset $\mathcal{D}$ is a finite collection of input/label pairs $\mathcal{D} = \{(x_1, y_1), \ldots..(x_N, y_N)\}$. The definition of DP relies on the notion of neighboring datasets, i.e datasets that vary by at most one example. We highlight below the central tools related to the field, inspired from Dwork et al. (2014).

**Definition 1** (($\epsilon, \delta$)-Approximate Differential Privacy). *Two datasets $\mathcal{D}$ and $\mathcal{D}'$ are said to be neighboring for the "add/remove-one" relation if they differ by exactly one sample: $|\mathcal{D}' \ominus \mathcal{D}| = 1$ where $\ominus$ denotes the symmetric difference between sets. Let $\epsilon$ and $\delta$ be two non-negative scalars. An*

---

[*]Equal contribution.

[†]IRIT, Université Paul Sabatier, Toulouse.

[‡]IRT Saint Exupéry, Toulouse.

[§]Inria, Université de Montpellier.

```python
model = DP_Sequential([ # step 1: use DP_Sequential to build a model
        # step 2: add Lipschitz layers of known sensitivity
        DP_BoundedInput(input_shape=(28, 28, 1), upper_bound=20.),
        DP_SpectralConv2D(filters=16, kernel_size=3, use_bias=False),
        DP_GroupSort(2),
        DP_Flatten(),
        DP_SpectralDense(1)],
    dp_parameters=dp_parameters,
    dataset_metadata=dataset_metadata,
) # step 4: compile the model, and choose any first order optimizer
model.compile(loss=DP_TauBCE(tau=20.), optimizer=Adam(1e-3))
model.fit( # step 5: train the model and measure the DP guarantees
    train_dataset, validation_data=val_dataset,
    epochs=num_epochs, callbacks=[DP_Accountant()]
)
```

Figure 1: **An example of usage of our framework** `lip-dp`, illustrating how to create a small Lipschitz VGG and how to train it under $(\epsilon, \delta)$-DP guarantees while reporting $(\epsilon, \delta)$ values.

*algorithm $\mathcal{A}$ is $(\epsilon, \delta)$-DP if for any two neighboring datasets $\mathcal{D}$ and $\mathcal{D}'$, and for any $S \subseteq range(\mathcal{A})$:*

$$\mathbb{P}[\mathcal{A}(\mathcal{D}) \in S] \leq e^{\epsilon} \times \mathbb{P}[\mathcal{A}(\mathcal{D}') \in S] + \delta. \tag{1}$$

A popular rule of thumb suggests using $\epsilon \leq 10$ and $\delta < \frac{1}{N}$ with $N$ the number of records (Ponomareva et al., 2023) for mild guarantees. In practice, most classic algorithmic procedures (called *queries* in this context) do not readily fulfill the definition for useful values of $(\epsilon, \delta)$: in particular, randomization is mandatory. A general recipe to make a query differentially private is to compute its *sensitivity* $\Delta$, and to perturb its output by adding a Gaussian noise of predefined variance $\zeta^2 = \Delta^2 \sigma^2$, where the $(\epsilon, \delta)$ guarantees depend on $\sigma$, yielding what is called a *Gaussian mechanism* (Dwork et al., 2006).

**Definition 2** ($l_2$-sensitivity). *Let $\mathcal{M}$ be a query mapping from the space of the datasets to $\mathbb{R}^p$. Let $\mathcal{N}$ be the set of all possible pairs of neighboring datasets $\mathcal{D}, \mathcal{D}'$. The $l_2$ sensitivity of $\mathcal{M}$ is defined by:*

$$\Delta(\mathcal{M}) = \sup_{\mathcal{D}, \mathcal{D}' \in \mathcal{N}} \|\mathcal{M}(D) - \mathcal{M}(D')\|_2. \tag{2}$$

This randomization comes at the expense of "utility", i.e the usefulness of the output for downstream tasks (Alvim et al., 2012). The goal is then to strike a balance between privacy and utility, ensuring that the released information remains useful and informative for the intended purpose while minimizing the risk of privacy breaches. The privacy/utility trade-off yields a Pareto front, materialized by plotting $\epsilon$ against a measurement of utility, such as validation accuracy for a classification task.

**Differentially Private SGD.** The SGD algorithm consists of a sequence of queries that (i) take the dataset in input, sample a minibatch from it, and return the gradient of the loss evaluated on the minibatch, before (ii) performing a descent step following the gradient direction. In "add-remove" neighboring relations, if the gradients are bounded by $K > 0$, the sensitivity of the gradients averaged on a minibatch of size $b$ is $\Delta = K/b$. DP-SGD (Abadi et al., 2016) makes each of these queries private by resorting to the Gaussian mechanism. Crucially, the algorithm requires a bound on gradient norms $\|\nabla_{\theta}\mathcal{L}(\hat{y}, y)\|_2 \leq C$. This upper bound on gradient norms is generally unknown in advance, which leads practitioners to clip it to $C > 0$, in order to bound the sensitivity manually. Unfortunately, this creates a number of issues: **1.** Hyper-parameter search on the broad-range clipping value $C$ is required to train models with good privacy/utility trade-offs (Papernot & Steinke, 2022), **2.** The computation of per-sample gradients is expensive: DP-SGD is usually slower and consumes more memory than vanilla SGD, in particular for the large batch sizes often used in private training (Lee & Kifer, 2021), **3.** Clipping the per-sample gradients biases their average (Chen et al., 2020). This is problematic as the average direction is mainly driven by misclassified examples.

**An unexplored approach: Lipschitz constrained networks.** To avoid these issues, we propose to train neural networks for which the parameter-wise gradients are provably and analytically bounded during the whole training procedure, in order to get rid of the clipping process. This allows for efficient training of models without the need for tedious hyper-parameter optimization. The main reason why this approach has not been experimented much in the past is that upper bounding the

gradient of neural networks is often intractable. However, by leveraging the literature on Lipschitz constrained networks introduced by Anil et al. (2019), we show that these networks have computable bounds for their gradient's norm. This yields tight bounds on the sensitivity of SGD steps, making their transformation into Gaussian mechanisms inexpensive - hence the name **Clipless DP-SGD**. Thus, our approach is qualitatively different from methods that replace clipping with re-normalization (Bu et al., 2022b; Yang et al., 2022), or from those that aim to reduce its time and memory footprint (Li et al., 2021; Bu et al., 2022a; He et al., 2022; Bu et al., 2023). A comparison is made in Appendix A.5.

The Lipschitz constant of a function and the norm of its gradient are two sides of the same coin. The function $f : \mathbb{R}^m \to \mathbb{R}^n$ is said $\ell$-Lipschitz for $l_2$ norm if for every $x, y \in \mathbb{R}^m$ we have $\|f(x) - f(y)\|_2 \leq \ell \|x - y\|_2$. Per Rademacher's theorem Simon et al. (1983), its gradient is bounded: $\|\nabla_x f\| \leq \ell$. Reciprocally, continuous functions with gradient bounded by $\ell$ are $\ell$-Lipschitz.

A feedforward neural network of depth $D$, with input space $\mathcal{X} \subset \mathbb{R}^n$, output space $\mathcal{Y} \subset \mathbb{R}^K$ (e.g logits), and parameter space $\Theta \subset \mathbb{R}^p$, is a parameterized function $f : \Theta \times \mathcal{X} \to \mathcal{Y}$ defined by the sequential composition of layers $f_1 \ldots f_D$:

$$f(\theta, x) := (f_D(\theta_D) \circ \ldots \circ f_2(\theta_2) \circ f_1(\theta_1))(x). \tag{3}$$

The parameters of the layers are denoted by $\theta = (\theta_d)_{1 \leq d \leq D} \in \Theta$. For affine layers, it corresponds to bias and weight matrix $\theta_d = (W_d, b_d)$. For activation functions, there are no parameters: $\theta_d = \varnothing$. Feedforward networks include notably Fully Connected networks, CNN, Resnets, or MLP mixers.

**Definition 3** (Lipschitz feed-forward neural network). *Lipschitz networks are feedforward networks, with the additional constraint that each layer $x_d \mapsto f_d(\theta_d, x_d) := y_d$ is $\ell_d$-Lipschitz for all $\theta_d$. Consequently, the function $x \mapsto f(\theta, x)$ is $\ell$-Lipschitz with $\ell = \ell_1 \times \ldots \times \ell_D$ for all $\theta \in \Theta$.*

The literature has predominantly focused on investigating the control of Lipschitzness with respect to the inputs (i.e bounding $\nabla_x f$), primarily motivated by concerns of robustness (Szegedy et al., 2014; Li et al., 2019a; Fazlyab et al., 2019), or improved generalization (Bartlett et al., 2017; Béthune et al., 2022). However, in this work, we will demonstrate that it is also possible to control Lipschitzness with respect to parameters (i.e bounding $\nabla_\theta f$), which is essential for ensuring privacy. Our first contribution will point out the tight link that exists between those two quantities. The closest work to ours is Shavit & Gjura (2019), where a Lipschitz network is used as a general function approximator of bounded sensitivity $\|\nabla_x f(\theta, x)\|_2$, but to this day the sensitivity of the gradient query $x \mapsto \nabla_\theta f(\theta, x)$ itself remains largely unexplored. The idea of using automatic differentiation to compute sensitivity bounds has also been discussed in Ziller et al. (2021) and Usynin et al. (2021).

**Contributions**. While the properties of Lipschitz constrained networks regarding their inputs are well explored, the properties with respect to their parameters remain non-trivial. This work provides a first step to fill this gap: our analysis shows that under appropriate architectural constraints, a $l$-Lipschitz network has a tractable, finite Lipschitz constant with respect to its parameters, which allows for easy estimation of the sensitivity of the gradient computation queries. Our contributions are the following:

1. We extend the field of applications of Lipschitz constrained neural networks. We extend the framework to **compute the Lipschitzness with respect to the parameters**. This general framework allows to track *layer-wise* sensitivities that depends on the loss and the model's structure. This is exposed in Section 2. We show that SGD training of deep neural networks can be achieved **without per-sample gradient clipping** using Lipschitz constrained layers.
2. We establish connections between Gradient Norm Preserving (GNP) networks and improved privacy/utility trade-offs (Section 3.1). To the best of our knowledge, **we are the first ones to produce neural networks benefiting from both Lipschitz-based robustness certificates and privacy guarantees**.
3. Finally, a **Python package** companions the project, with pre-computed Lipschitz constants for each loss and each layer type. This is exposed in Section 3.2. It covers widely used architectures, including VGG, ResNets or MLP Mixers. Our package enables the use of larger networks and larger batch sizes, as illustrated by our experiments in Section 4.

## 2 CLIPLESS DP-SGD WITH $\ell$-LIPSCHITZ NETWORKS

Our framework relies on the computation of the maximum gradient norm of a network w.r.t its parameters to obtain a *per-layer* sensitivity $\Delta_d$. It is based on the recursive formulation of the chain rule involved in backpropagation and requires some natural assumptions that we highlight below.

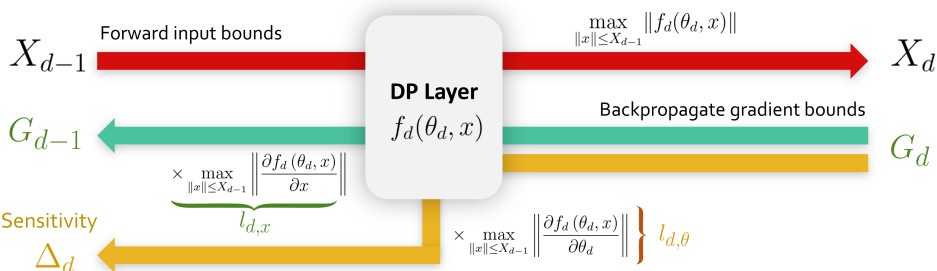

Figure 2: **Backpropagation for bounds** (Algorithm 1) computes the per-layer sensitivity $\Delta_d$.

**Requirement 1** (Lipschitz loss.). *The loss function $\hat{y} \mapsto \mathcal{L}(\hat{y}, y)$ must be $L$-Lipschitz with respect to the logits $\hat{y}$ for all ground truths $y \in \mathcal{Y}$. This is notably the case of Categorical Softmax-Crossentropy.*

The Lipschitz constants of common supervised losses has been computed and reported in the appendix.

**Requirement 2** (Bounded input). *There exists $X_0 > 0$ such that for all $x \in \mathcal{X}$ we have $\|x\| \leq X_0$.*

There exist numerous approaches for the parametrization of Lipschitz networks, such as differentiable re-parametrization (aka "trivialization") like Miyato et al. (2018); Anil et al. (2019), optimization over matrix manifolds (Absil et al., 2008), direct parametrization (Meunier et al., 2022; Wang & Manchester, 2023), or projections (Arjovsky et al., 2017). We can distinguish two strategies:

1. With a differentiable reparametrization $\Pi : \mathbb{R}^p \to \Theta$ where $\tilde{\theta} = \Pi(\theta)$: the weights $\tilde{\theta}$ are used during the forward pass, but the gradients are back-propagated to $\theta$ through $\Pi$. This turns the training into an unconstrained optimization problem on the landscape of $\mathcal{L} \circ f \circ \Pi$.
2. With a suitable projection operator $\Pi : \mathbb{R}^p \to \Theta$: this is the celebrated Projected Gradient Descent (PGD) algorithm (Bubeck et al., 2015) applied on the landscape of $\mathcal{L} \circ f$.

Option 1 requires the analysis of the Lipschitz constant of $\Pi$. If $\Theta$ is convex, then $\Pi$ is 1-Lipschitz w.r.t the projection norm, otherwise the Lipschitz constant is generally unknown. For simplicity, option 2 will be the focus of our work.

**Requirement 3** (Lipschitz projection). *The Lipschitz constraints must be enforced with a projection operator $\Pi : \mathbb{R}^p \to \Theta$. This corresponds to Tensorflow `constraints` and Pytorch `hooks`. Projection is a post-processing (Dwork et al., 2006) of private data: it induces no privacy leakage.*

To compute the per-layer sensitivities, our framework mimics the backpropagation algorithm, where *Vector-Jacobian* products (VJP) are replaced by *Scalar-Scalar* products of element-wise bounds. For an arbitrary layer $x_d \mapsto f_d(\theta_d, x_d) := y_d$ the operation is sketched below, as a simple consequence of Cauchy-Schwartz inequality:

$$\underbrace{\nabla_{x_d}\mathcal{L} := (\nabla_{y_d}\mathcal{L})\frac{\partial f_d}{\partial x_d}}_{\text{Vector-Jacobian product: backpropagate gradients}} \implies \underbrace{\|\nabla_{x_d}\mathcal{L}\|_2 \leq \|\nabla_{y_d}\mathcal{L}\|_2 \times \left\|\frac{\partial f_d}{\partial x_d}\right\|_2}_{\text{Scalar-Scalar product: backpropagate bounds}}. \tag{4}$$

The notation $\|\cdot\|_2$ must be understood as the spectral norm for Jacobian matrices, and the Euclidean norm for gradient vectors. The scalar-scalar product is inexpensive. For Lipschitz layers, the spectral norm of the Jacobian $\|\frac{\partial f}{\partial x}\|$ is kept constant during training with projection operator $\Pi$. The bound of the gradient with respect to the parameters takes a simple form:

$$\|\nabla_{\theta_d}\mathcal{L}\|_2 \leq \|\nabla_{y_d}\mathcal{L}\|_2 \times \left\|\frac{\partial f_d}{\partial \theta_d}\right\|_2. \tag{5}$$

This term can be analytically bounded, as exposed in the following section.

## 2.1 BACKPROPAGATION FOR BOUNDS

The pseudo-code of **Clipless DP-SGD** is sketched in Algorithm 2. The algorithm avoids per-sample clipping by computing a *per-layer* bound on the element-wise gradient norm. The computation of this

*per-layer* bound is described by Algorithm 1 (graphically explained in Figure 2). Crucially, it requires to compute the spectral norm of the Jacobian of each layer with respect to input and parameters.

**Input bound propagation (line 2).** We compute $X_d = \max_{\|x\| \leq X_{d-1}} \|f_d(x)\|_2$. For activation functions it depends on their range. For linear layers, it depends on the spectral norm of the operator itself. This quantity can be computed through the SVD or Power Iteration (Trefethen & Bau, 2022; Miyato et al., 2018), and constrained during training using projection operator $\Pi$. In particular, it covers the case of convolutions, for which tight bounds are known (Singla & Feizi, 2021a). For affine layers, it additionally depends on the magnitude of the bias $\|b_d\|$.

**Remark 1** (Tighter bounds in literature). *Exact estimation of the Lipschitz bound is notoriously an NP-hard problem, as proven by (Virmaux & Scaman, 2018). Algorithm 2 can be seen as an extension of their AutoLip algorithm to the case of parameters, while their work focused on Lipschitzness with respect to the input. Although libraries such as Decomon (Airbus, 2023) or auto-LiRPA (Xu et al., 2020) provide tighter bounds for $X_d$ via linear relaxations (Singh et al., 2019; Zhang et al., 2018), our approach is capable of delivering practically tighter bounds than worst-case scenarios thanks to the projection operator $\Pi$, while also being significantly less computationally expensive. Moreover, hybridizing our method with scalable certification methods can be a path for future extensions.*

**Computing maximum gradient norm (line 6).** We now present how to bound the Jacobian $\frac{\partial f_d(\theta_d, x)}{\partial \theta_d}$. In neural networks, the parameterized layers $f(\theta, x)$ (fully connected, convolutions) are bilinear operators. Hence, we typically obtain bounds of the form:

$$\left\| \frac{\partial f_d(\theta_d, x)}{\partial \theta_d} \right\|_2 \leq K(f_d, \theta_d)\|x\|_2 \leq K(f_d, \theta_d)X_{d-1}, \tag{6}$$

where $K(f_d, \theta_d)$ is a constant that depends on the nature of the operator. $X_{d-1}$ is obtained in line 2 with input bound propagation. Values of $K(f_d, \theta_d)$ for popular layers are reported in the appendix.

**Backpropagate cotangent vector bounds (line 7).** Finally, we bound the Jacobian $\frac{\partial f_d(\theta_d, x)}{\partial x}$. For activation functions, this value can be hard-coded, while for affine layers it is the spectral norm of the linear operator. Like before, this value is enforced by the projection operator $\Pi$.

---

**Algorithm 1 Backpropagation for Bounds**$(f, X)$

---

**Input**: Feed-forward architecture $f(\theta, \cdot) = f_D(\theta_D, \cdot) \circ \ldots \circ f_1(\theta_1, \cdot)$
**Input**: Weights $\theta = (\theta_1, \theta_2, \ldots \theta_D)$, input bound $X_0$

1: **for all** layers $1 \leq d \leq D$ **do**
2:     $X_d \leftarrow \max\limits_{\|x\| \leq X_{d-1}} \|f_d(\theta_d, x)\|_2$.         ▷ Input bounds propagation
3: **end for**
4: $G \leftarrow L/b$.         ▷ Lipschitz constant of the (averaged) loss for batchsize b
5: **for all** layers $D \geq d \geq 1$ **do**
6:     $\Delta_d \leftarrow G \max\limits_{\|x\| \leq X_{d-1}} \|\frac{\partial f_d(\theta_d, x)}{\partial \theta_d}\|_2$.     ▷ Compute sensitivity from gradient norm
7:     $G \leftarrow G \max\limits_{\|x\| \leq X_{d-1}} \|\frac{\partial f_d(\theta_d, x)}{\partial x}\|_2 = Gl_d$.     ▷ Backpropagate cotangent vector bounds
8: **end for**
9: **return** sensitivities $\Delta_1, \Delta_2 \ldots, \Delta_D$

---

**Privacy accounting for Clipless DP-SGD.** We keep track of $(\epsilon, \delta)$-DP values with a *privacy accountant* (Abadi et al., 2016), by composing different mechanisms. For a dataset with $N$ records and a batch size $b$, it relies on two parameters: the sampling ratio $p = \frac{b}{N}$ and the "noise multiplier" $\sigma$ defined as the ratio between effective noise strength $\zeta$ and sensitivity $\Delta$. We propose two strategies to keep track of $(\epsilon, \delta)$ values as the training progresses, based on either the "per-layer" sensitivities $\Delta_d$ (composition of Gaussian mechanisms), or by aggregating them into a "global" sensitivity $\Delta = \sqrt{\sum_d \Delta_d^2}$ (single isotropic Gaussian mechanism). We rely on the `autodp`[1] library as it uses the Rényi Differential Privacy (RDP) adaptive composition theorem (Mironov, 2017), that ensures tighter bounds than naive DP composition. Following standard practices of the community (Ponomareva

---

[1]`https://github.com/yuxiangw/autodp` distributed under Apache License 2.0.

**Algorithm 2 Clipless DP-SGD** with **per-layer** sensitivity accounting

---

**Input**: Feed-forward architecture $f(\theta, \cdot) = f_D(\theta_D, \cdot) \circ \ldots \circ f_1(\theta_1, \cdot)$
**Input**: Initial weights $\theta = (\theta_1, \theta_1, \ldots \theta_D)$, learning rate $\eta$, noise multiplier $\sigma$.
 1: **repeat**
 2:     $\Delta_1, \Delta_2 \ldots \Delta_D \leftarrow$ **Backpropagation for Bounds**$(f, X)$.
 3:     Sample a batch $\mathcal{B} = \{(x_1, y_1), (x_2, y_2), \ldots, (x_b, y_b)\}$.
 4:     Compute the averaged gradient for each layer $d$: $g_d := \frac{1}{b} \sum_{i=1}^{b} \nabla_{\theta_d} \mathcal{L}(f(\theta, x_i), y_i))$.
 5:     Sample per-layer noise: $\zeta_d \sim \mathcal{N}(\mathbf{0}, \sigma\Delta_d)$.
 6:     Perform perturbed gradient step: $\theta_d \leftarrow \theta_d - \eta(g_d + \zeta_d)$.
 7:     Enforce Lipschitz constraint with projection: $\theta_d \leftarrow \Pi(\theta_d)$.
 8:     Compute new $(\epsilon, \delta)$-DP guarantees with privacy accountant.
 9: **until** privacy budget $(\epsilon, \delta)$ has been reached.

---

et al., 2023), we used *sampling without replacement* at each epoch (by shuffling examples), but we reported $\epsilon$ assuming *Poisson sampling* to benefit from privacy amplification (Balle et al., 2018).

## 3   Signal-to-noise ratio analysis

We discuss how the tightness of the bound provided by Algorithm 1 can be controlled.

### 3.1   Theoretical analysis of Clipless DP-SGD

In some cases we can manually derive the bounds across diverse configurations.

**Theorem (informal) 1. Gradient Norm of Lipschitz Networks.** *Assume that every layer $f_d$ is $K$-Lipschitz, i.e $l_1 = \cdots = l_D = K$. Assume that every bias is bounded by $B$. We further assume that each activation is centered at zero (i.e $f_d(\mathbf{0}) = \mathbf{0}$, like ReLU, GroupSort...).Then the global upper bound of Algorithm 2 can be computed analytically, as follows:*

*1. If $K < 1$ we have:* $\|\nabla_\theta \mathcal{L}(f(\theta, x), y)\|_2 = \mathcal{O}\left(L\left(K^D(X_0 + B) + 1\right)\right)$.
*Due to the $K^D \ll 1$ term this corresponds to a vanishing gradient phenomenon (Pascanu et al., 2013). The output of the network is essentially independent of its input, and training is nearly impossible.*

*2. If $K > 1$ we have:* $\|\nabla_\theta \mathcal{L}(f(\theta, x), y)\|_2 = \mathcal{O}\left(LK^D(X_0 + B + 1)\right)$.
*Due to the $K^D \gg 1$ term this corresponds to an exploding gradient phenomenon (Bengio et al., 1994). The upper bound becomes vacuous for deep networks: the added noise $\zeta$ will be too high.*

*3. If $K = 1$ we have:* $\|\nabla_\theta \mathcal{L}(f(\theta, x), y)\|_2 = \mathcal{O}\left(L\left(\sqrt{D} + X_0\sqrt{D} + \sqrt{BX_0}D + BD^{3/2}\right)\right)$,
*which for linear layers without biases further simplify to $\mathcal{O}(L\sqrt{D}(1 + X_0))$.*

The formal statement can be found in appendix. We see that most favorable bounds are achieved by 1-Lipschitz neural networks with 1-Lipschitz layers. In classification tasks, they are as expressive as conventional networks (Béthune et al., 2022). Hence, this choice of architecture is not at the expense of utility. Moreover an accuracy/robustness trade-off exists, determined by the choice of loss function (Béthune et al., 2022). However, setting $K = 1$ merely ensures that $\|\nabla_x f\| \leq 1$, and in the worst-case scenario we could have $\|\nabla_x f\| \ll 1$ almost everywhere. This results in a situation where the bound of case 3 in Theorem 1 is not tight, leading to an underfitting regime as in the case $K < 1$. With Gradient Norm Preserving (GNP) networks, we expect to mitigate this issue.

**Controlling $K$ with Gradient Norm Preserving (GNP) networks.**   GNP (Li et al., 2019a) networks are 1-Lipschitz neural networks with the additional constraint that the Jacobian of layers consists of orthogonal matrices. They fulfill the Eikonal equation $\left\|\frac{\partial f_d(\theta_d, x_d)}{\partial x_d}\right\|_2 = 1$ for any intermediate activation $f_d(\theta_d, x_d)$. As a consequence, the gradient of the loss with respect to the parameters

is bounded by

$$\|\nabla_{\theta_d}\mathcal{L}\| \leq \|\nabla_{y_D}\mathcal{L}\| \times \left\|\prod_{d<i\leq D}\frac{\partial f_i(\theta_i, x_i)}{\partial x_i}\right\| \times \left\|\frac{\partial f_d(\theta_d, x_d)}{\partial \theta_d}\right\| = \|\nabla_{y_D}\mathcal{L}\| \times \left\|\frac{\partial f_d(\theta_d, x_d)}{\partial \theta_d}\right\|, \quad (7)$$

which for weight matrices $W_d$ further simplifies to $\|\nabla_{W_d}\mathcal{L}\| \leq \|\nabla_{y_D}\mathcal{L}\| \times \|f_{d-1}(\theta_{d-1}, x_{d-1})\|$. We see that this upper bound crucially depends on two terms than can be analyzed separately. On the one hand, $\|f_{d-1}(\theta_{d-1}, x_{d-1})\|$ depends on the scale of the input. On the other, $\|\nabla_{y_D}\mathcal{L}\|$ depends on the loss, the predictions and the training stage. We show below how to intervene on these two quantities.

**Controlling $X_0$ with input pre-processing.** The weight gradient norm $\|\nabla_{W_d}\mathcal{L}\|$ indirectly depends on the norm of the inputs. Multiple strategies are available to keep this norm under control: projection onto the ball ("norm clipping"), or projection onto the sphere ("normalization"). In the domain of natural images, this result sheds light on the importance of color space: RGB, HSV, etc. Empirically, a narrower distribution of input norms would make up for tighter gradient norm bounds.

**Controlling $L$ with the hybrid approach: loss gradient clipping** As training progresses, the magnitude of $\|\nabla_f\mathcal{L}\|$ tends to diminish when approaching local minima, falling below the upper bound and diminishing the signal to noise ratio. Fortunately, for Lipschitz constrained networks, the norm of the elementwise-gradient remains lower bounded throughout (Béthune et al., 2022). Since the noise amplitude only depends on the architecture and the loss, and remains fixed during training, the loss with the best signal-to-noise ratio would be a loss whose gradient norm w.r.t the logits remains constant during training. For the binary classification case, with labels $y \in \{-1, +1\}$, this yields the loss $\mathcal{L}_{KR}(\hat{y}, y) = -y\hat{y}$, that arises in Kantorovich-Rubinstein duality (Villani, 2008):

$$\mathcal{W}_1(P, Q) := \frac{1}{\ell}\inf_{f \in \ell\text{-Lip}(\mathcal{D}, \mathbb{R})}\mathbb{E}_{(x,y)\sim\mathcal{D}}[\mathcal{L}_{KR}(f(x), y)], \quad (8)$$

where $\mathcal{W}_1(P, Q)$ is the Wasserstein-1 distance between the two classes $P$ and $Q$.

Another way to ensure high signal-to-noise ratio is to diminish the noise and clip the gradients of the loss w.r.t the logits. We emphasize that *this is different from the clipping of the "parameter gradient"* $\nabla_\theta\mathcal{L}$ done in DP-SGD. Here, *any intermediate gradient $\nabla_{f_d}\mathcal{L}$ can be clipped during backpropagation.* This can be achieved with a special *"clipping layer"* that behaves like the identity function at the forward pass, and clips the gradient during the backward pass. In DP-SGD the clipping is applied on the element-wise gradient $\nabla_{W_d}\mathcal{L}$ of size $b \times h^2$ for matrix weight $W_d \in \mathbb{R}^{h\times h}$ and batch size $b$, and clipping it can cause memory issues or slowdowns (Lee & Kifer, 2021). In our case, $\nabla_{y_D}\mathcal{L}$ is of size $b \times h$: this is far smaller, especially for the last layer. Moreover, this clipping is compatible with the adaptive clipping introduced by (Andrew et al., 2021): the quantiles of $\|\nabla_{y_D}\mathcal{L}\|$ can be privately estimated for a small privacy budget. This allows to effectively reduce the noise while ensuring that most of the gradients remain unbiased. Furthermore, this bias of this clipping can be characterized.

**Proposition (informal) 1** (Bias of loss gradient clipping in binary classification tasks). *Let $\mathcal{L}_{BCE}$ be the binary cross-entropy loss, with sigmoid activation. Assume that the loss gradient (w.r.t the logits) $\nabla_{\hat{y}}\mathbb{E}_{\mathcal{D}}[\mathcal{L}_{BCE}(\hat{y}, y)]$ is clipped to norm at most $C > 0$. Then there exists $C' > 0$ such that for all $C \leq C'$ a gradient descent step with the clipped gradient is identical in direction to the gradient descent step obtained from the loss $\mathcal{L}_{KR}(\hat{y}, y) = -\hat{y}y$.*

As observed in Serrurier et al. (2021) and Béthune et al. (2022) this descent direction yields classifiers with high certifiable robustness, but lower clean accuracy. Therefore, in practice, we set the adaptive clipping threshold at not less than the 90%-th quantile to mitigate the bias and avoid utility drop.

## 3.2 LIP-DP LIBRARY

To foster and spread accessibility, we provide an open source TensorFlow library for Clipless DP-SGD training[2] , named `lip-dp`, with Keras API. The code is available at `https://github.com/Algue-Rythme/lip-dp`. Its usage is illustrated in Figure 1. We rely on the `deel-lip`[3] library Serrurier et al. (2021) to enforce Lipschitz constraints in practice, with Reshaped Kernel

---

[2]`https://anonymous.4open.science/r/lipdp-5EA1` distributed under MIT License.
[3]`https://github.com/deel-ai/deel-lip` distributed under MIT License.

| Dataset | Size | Dim. | $\delta$ | Validation AUROC ↑ | |
|---|---|---|---|---|---|
| | | | | DP-SGD | Clipless DP-SGD |
| ALOI | 39,627 | 27 | $10^{-5}$ | **56.5** | 56.2 |
| campaign | 32,950 | 62 | $10^{-5}$ | **90.0** | 82.2 |
| celeba | 162,079 | 39 | $10^{-6}$ | **96.6** | 96.5 |
| census | 239,428 | 500 | $10^{-6}$ | **93.3** | 92.5 |
| donors | 495,460 | 10 | $10^{-6}$ | **100.0** | **100.0** |
| magic | 15,216 | 10 | $10^{-5}$ | **90.7** | 89.7 |
| shuttle | 39,277 | 9 | $10^{-5}$ | 98.3 | **99.4** |
| skin | 196,045 | 3 | $10^{-6}$ | **100.0** | 99.8 |
| yeast | 1,187 | 8 | $10^{-4}$ | 66.8 | **75.1** |

(a) **Best AUROC values (in %) for models trained under** $(\epsilon, \delta)$**-DP privacy with** $\epsilon = 1$, on binary classification tasks of tabular data from Adbench datasets Han et al. (2022). We use a random stratified split into train (80%) / validation (20%).

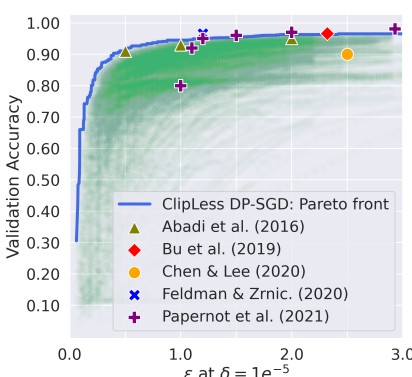

(b) **Pareto front on Mnist.** Each green dot corresponds to a (**val. acc**, $\epsilon$) pair from an epoch.

Figure 3: **Comparison of the utility of DP-SGD and Clipless DP-SGD** on tabular and image data.

Orthogonalization (RKO) for fast and near-orthogonal convolutions (Li et al., 2019b). The Lipschitz constraint is enforced by using activations with known Lipschitz constant $l_d$ (most activations fulfill this condition), and by applying a projection $\Pi : \mathbb{R}^p \to \Theta$ on the weights of affine layers. These constraints correspond to spectrally normalized matrices Yoshida & Miyato (2017); Bartlett et al. (2017), since for affine layers we have $\ell_d = \|W_d\|_2 := \max_{\|x\| \leq 1} \|W_d x\|_2$ and therefore $\Theta = \{\|W_d\|_2 \leq \ell_q\}$. We rely on Power Iteration for fast computation of the spectral norm. The leading eigenvector is cached from a step to another to speed-up computations. The seminal work of Anil et al. (2019) proved that universal approximation in the set of $\ell$-Lipschitz functions was achievable by this family of architectures. In practice, GNP networks are parametrized with GroupSort activation Anil et al. (2019); Tanielian & Biau (2021), Householder activation Mhammedi et al. (2017), and orthogonal weight matrices Li et al. (2019a;b).

**Remark 2** (GNP networks limitations). *Strict orthogonality is challenging to enforce, especially for convolutions for which it is still an active research area (see Trockman & Kolter (2021); Singla & Feizi (2021b); Achour et al. (2022); Singla & Feizi (2022); Xu et al. (2022) and references therein). Our line of work traces an additional motivation for the development of GNP and the bounds will strengthen as the field progresses. Concurrent approaches based on regularization (Gulrajani et al., 2017; Cisse et al., 2017; Gouk et al., 2021)) fail to produce formal guarantees.*

## 4 EXPERIMENTAL RESULTS

We validate our implementation with a speed benchmark against competing approaches, and we present the privacy/utility Pareto fronts that can be obtained with GNP networks.

### 4.1 EVALUATION OF PRIVACY, ACCURACY AND ROBUSTNESS

For the comparisons, we leverage the DP-SGD implementation from Opacus. We perform a search over a broad range of hyper-parameter values: the configuration is given in Appendix D. We ignore the privacy loss that may be induced by this hyper-parameter search, which is a limitation per recent studies like Papernot & Steinke (2022) or Ding & Wu (2022).

**Accuracy and Privacy.** We validate the performance of our approach on tabular data from Adbench suite (Han et al., 2022) using a MLP, and report the result in Table 3a. For MNIST (Fig. 3b we use a Lipschitz LeNet-like architecture.

**Robustness and Privacy.** One of the most prominent advantage of Lipchitz networks is their ability to provide robustness certificates against adversarial attacks, and was the primary motivation for their development (Szegedy et al., 2014; Li et al., 2019a; Fazlyab et al., 2019). For a $\ell$-Lipschitz classifier $f$, with predictions $\bar{k} := \arg\max_k f_k(x)$ the decision is invariant under perturbation of norms smaller than $\frac{1}{\ell\sqrt{2}}(f_{\bar{k}}(x) - \arg\max_{i \neq \bar{k}} f_i(x))$. Therefore, for each perturbation radius $r$ we

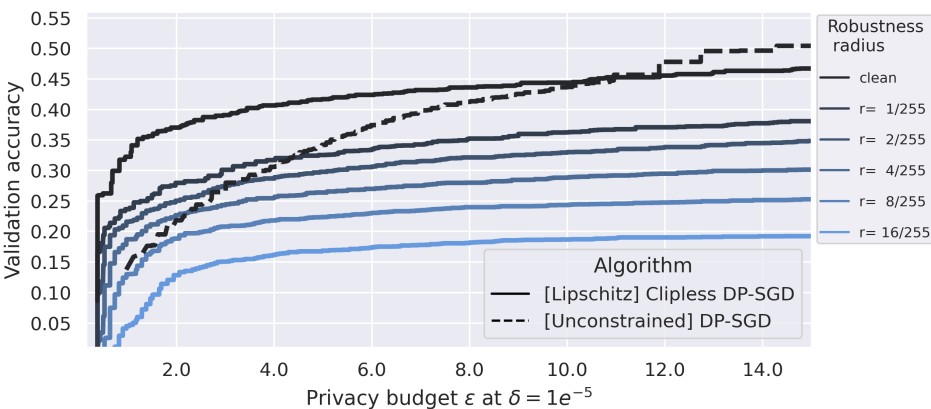

Figure 4: **Privacy/accuracy/robustness trade-off on Cifar-10:** We report the pareto front of robustness certificates at different radii $r$ for Lipschitz constrained networks, while unconstrained networks cannot produce robustness certificates. Models are trained in an "out of the box setting": no pre-training, no data augmentation and no handcrafted features. To ensure a fair comparison between algorithms, we perform 30 repetitions with a Bayesian optimizer to select the best hyper-parameters.

can measure the accuracy of the perturbed network, and we report these results in Fig. 4. The computation of the certificates is straightforward, while methods applicable to conventional networks (like randomized smoothing (Cohen et al., 2019) or interval propagation, see remark 1), are expensive. This suggests that robust decisions and privacy are not necessarily antipodal objectives, contrary to what was observed in Song et al. (2019). The work of Wu et al. (2023) also study the link between certified robustness and privacy, albeit through the lens of adversarial training (Zhang et al., 2019).

### 4.2 SPEED AND MEMORY CONSUMPTION

We benchmarked the median runtime per batch of vanilla DP-SGD against the one of Clipless DP-SGD, on a CNN architecture and its Lipschitz equivalent respectively. The experiment was run on a GPU with 48GB video memory. We compare against the implementation of `tf_privacy`, `opacus` and `optax`. In order to allow a fair comparison, when evaluating Opacus, we reported the runtime with respect to the logical batch size, while capping the physical batch size to avoid Out Of Memory error (OOM).

An advantage of the projection $\Pi$ over per-sample gradient clipping is that its cost is independent of the batch size. Fig 5 validates that our method scales much better than vanilla DP-SGD, and is compatible with large batch sizes. It offers several advantages: firstly, a larger batch size contributes to a decrease of the sensitivity $\Delta \propto 1/b$, which diminishes the ratio between noise and gradient norm. Secondly, as the batch size $b$ increases, the variance decreases at the parametric rate $\mathcal{O}(\sqrt{b})$ (as demonstrated in appendix E.1), aligning with expectations. This observation does not apply to DP-SGD: clipping biases the direction of the average gradient, as noticed by Chen et al. (2020).

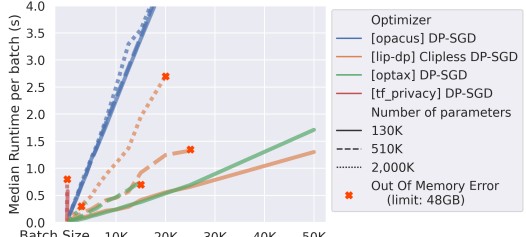

Figure 5: **Our approach outperforms concurrent frameworks in terms of runtime and memory:** we trained CNNs (ranging from 130K to 2M parameters) on CIFAR-10, and report the median batch processing time (including noise, and constraints application $\Pi$ or gradient clipping).

## 5 LIMITATIONS, FUTURE WORK AND BROADER IMPACT

Our framework offers a novel approach to address differentially private training, but also introduces new challenges. Clipless DP-SGD replaces a bias of optimizer with a bias in function space (as

discussed in Appendix A.4). We primary rely on GNP networks, where high performing architectures are quite different from the usual CNN architectures. This serves as **(1) a motivation to further develop Gradient Norm Preserving architectures**. As emphasized in Remark 2, we anticipate that progress in these areas will greatly enhance the effectiveness of our approach. Additionally, to meet requirement 3, we rely on projections, necessitating additional efforts to incorporate "trivializations" Trockman & Kolter (2021); Singla & Feizi (2021b). Finally, as mentioned in Remark 1, our propagation bound method can be refined. Comparison of Clipless DP-SGD with recent methods based on back-propagation, in particular Bu et al. (2023), is discussed in appendix A.5. Furthermore, the development of networks with known Lipschitz constant with respect to parameters is a question of independent interest, making `lip-dp` **(2) a useful tool for the study of the optimization dynamics** in Lipschitz constrained neural networks.

## CONTRIBUTIONS

The design and the theoretical analysis has been conducted by Louis Béthune and Thomas Masséna. The lip-dp package has been created by Louis Bethune, Thomas Masséna, and Thibaut Boissin. The experiments have been run by Louis Béthune, Thomas Masséna, Thibaut Boissin, Yannick Prudent, and Corentin Friedrich. Proofreading, writing, and feedback are a joint contribution of all authors.

## ACKNOWLEDGMENTS

Louis thanks Fabian Pedregosa for its detailed feedback. The authors thank Agustin Picard for its proofreading.

This work has benefited from the AI Interdisciplinary Institute ANITI, which is funded by the French "Investing for the Future – PIA3" program under the Grant agreement ANR-19-P3IA-0004. The authors gratefully acknowledge the support of the DEEL project.[4]

This work was supported by grant ANR-20-CE23-0015 (Project PRIDE) and the ANR 22-PECY-0002 IPOP (Interdisciplinary Project on Privacy) project of the Cybersecurity PEPR.

---

[4]`https://www.deel.ai/`

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

CONTENTS

# A  DEFINITIONS AND METHODS

## A.1  ADDITIONNAL BACKGROUND

The purpose of this appendix is to provide additionnal definitions and properties regarding Lipschitz Neural Networks, their possible GNP properties and Differential Privacy.

### A.1.1  LIPSCHITZ NEURAL NETWORKS BACKGROUND

For simplicity of the exposure, we will focus on feedforward neural networks with densely connected layers: the affine transformation takes the form of a matrix-vector product $h \mapsto Wh$. In Section C.2 we tackle the case of convolutions $h \mapsto \Psi * h$ with kernel $\Psi$.

**Definition 4** (Feedforward neural network). *A feedforward neural network of depth $T$, with input space $\mathcal{X} \subset \mathbb{R}^n$, and with parameter space $\Theta \subset \mathbb{R}^p$, is a parameterized function $f : \Theta \times \mathcal{X} \to \mathcal{Y}$ defined by the following recursion:*

$$
\begin{aligned}
h_0(x) &:= x, & z_t(x) &:= W_t h_{t-1}(x) + b_t, \\
h_t(x) &:= \sigma(z_t(x)), & f(\theta, x) &:= z_{T+1}(x).
\end{aligned}
\tag{9}
$$

*The set of parameters is denoted as $\theta = (W_t, b_t)_{1 \leq t \leq T+1}$, the output space as $\mathcal{Y} \subset \mathbb{R}^K$ (e.g logits), and the layer-wise activation as $\sigma : \mathbb{R}^n \to \mathbb{R}^n$.*

**Definition 5** (Lipschitz constant). *The function $f : \mathbb{R}^m \to \mathbb{R}^n$ is said $l$-Lipschitz for $l_2$ norm if for every $x, y \in \mathbb{R}^m$ we have:*

$$
\|f(x) - f(y)\|_2 \leq l\|x - y\|_2.
\tag{10}
$$

*Per Rademacher's theorem Simon et al. (1983), its gradient is bounded: $\|\nabla f\| \leq l$. Reciprocally, continuous functions gradient bounded by $l$ are $l$-Lipschitz.*

**Definition 6** (Lipschitz neural network). *A Lipschitz neural network is a feedforward neural network with the additional constraints:*

- *the activation function $\sigma$ is $S$-Lipschitz. This is a standard assumption, frequently fulfilled in practice.*

- *the affine functions $x \mapsto Wx + b$ are $U$-Lipschitz, i.e $\|W\|_2 \leq U$. This is achieved in practice with spectrally normalized matrices Yoshida & Miyato (2017) Bartlett et al. (2017). The feasible set is the ball $\{\|W\|_2 \leq U\}$ of radius $U$ (which is convex), or a subset of thereof (not necessarily convex).*

*As a result, the function $x \mapsto f(\theta, x)$ is $U(US)^T$-Lipschitz for all $\theta \in \Theta$.*

Two strategies are available to enforce Lipschitzness:

1. With a differentiable reparametrization $\Pi : \mathbb{R}^p \to \Theta$ where $\tilde{\theta} = \Pi(\theta)$: the weights $\tilde{\theta}$ are used during the forward pass, but the gradients are back-propagated to $\theta$ through $\Pi$. This turns the training into an unconstrained optimization problem on the landscape of $\mathcal{L} \circ f \circ \Pi$.

2. With a suitable projection operator $\Pi : \mathbb{R}^p \to \Theta$: this is the celebrated Projected Gradient Descent (PGD) algorithm Bubeck et al. (2015) applied on the landscape of $\mathcal{L} \circ f$.

For arbitrary re-parametrizations, option 1 can cause some difficulties: the Lipschiz constant of $\Pi$ is generally unknown. However, if $\Theta$ is convex then $\Pi$ is 1-Lipschitz (with respect to the norm chosen for the projection). To the contrary, option 2 elicits a broader set of feasible sets $\Theta$. For simplicity, option 2 will be the focus of our work.

### A.1.2  GRADIENT NORM PRESERVING NETWORKS

**Definition 7** (Gradient Norm Preserving Networks). *GNP networks are 1-Lipschitz neural networks with the additional constraint that the Jacobian of layers consists of orthogonal matrices:*

$$
\left(\frac{\partial f_d}{\partial x_d}\right)^T \left(\frac{\partial f_d}{\partial x_d}\right) = I.
\tag{11}
$$

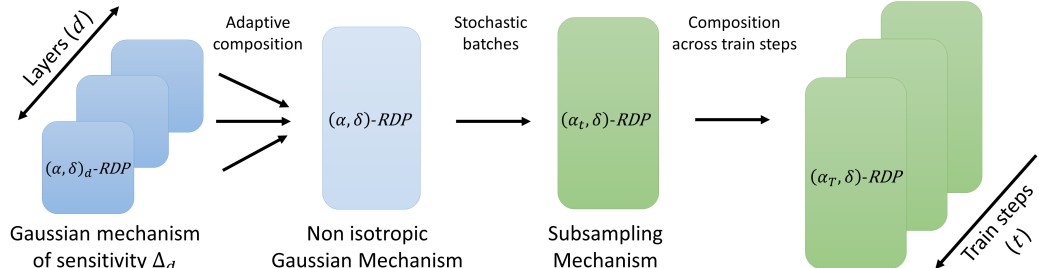

Figure 6: **Accountant for locally enforced differential privacy. (i)** The gradient query for each layer is made private using the Gaussian mechanism Dwork et al. (2014); **(ii)** their composition across the layers of the whole network can be seen as a non isotropic Gaussian mechanism, **(iii)** that benefits from amplification via sub-sampling Balle et al. (2018); **(iv)** the train steps are composed over the course of training.

*This is achieved with GroupSort activation Anil et al. (2019); Tanielian & Biau (2021), Householder activation Mhammedi et al. (2017), and orthogonal weight matrices Li et al. (2019a;b) or orthogonal convolutions (see Achour et al. (2022); Singla & Feizi (2022); Xu et al. (2022) and references therein). Without biases these networks are also norm preserving: $\|f(\theta, x)\| = \|x\|$.*

The set of orthogonal matrices (and its generalization the Stiefel manifold Absil et al. (2008)) is not convex, and not even connected. Hence projected gradient approaches are mandatory: for re-parametrization methods the Jacobian $\frac{\partial \Pi}{\partial \theta}$ may be unbounded which could have uncontrollable consequences on sensitivity.

### A.1.3 BACKGROUND ON DIFFERENTIAL PRIVACY

**Definition 8** (Neighboring datasets)**.** *A labelled dataset $\mathcal{D}$ is a finite collection of input/label pairs $\mathcal{D} = \{(x_1, y_1), (x_2, y_2), \ldots (x_N, y_N)\}$. Two datasets $\mathcal{D}$ and $\mathcal{D}'$ are said to be neighbouring if they differ by at most one sample: $\mathcal{D}' = \mathcal{D} \cup \{(x', y')\} - \{(x_i, y_i)\}$.*

The sensitivity is also referred as algorithmic stability Bousquet & Elisseeff (2002), or bounded differences property in other fields McDiarmid et al. (1989). We detail below the building of a Gaussian mechanism from an arbitrary query of known sensitivity.

**Definition 9** (Gaussian Mechanism)**.** *Let $f : \mathcal{D} \to \mathbb{R}^p$ be a query accessing the dataset of known $l_2$-sensitivity $\Delta(f)$, a Gaussian mechanism adds noise sampled from $\mathcal{N}(0, \sigma.\Delta(f))$ to the query $f$.*

**Property 1** (DP of Gaussian Mechanisms)**.** *Let $\mathcal{G}(f).$ be a Gaussian mechanism of $l_2$-sensitivity $S_2(f)$ adding the noise $\mathcal{N}(0, \sigma.S_2(f))$ to the query $f$. The DP guarantees of the mechanism are given by the following continuum: $\sigma = \sqrt{2.\log(1.25/\delta)}/\epsilon$.*

SGD is a composition of queries. Each of those query consists of sampling a minibatch from the dataset, and computing the gradient of the loss on the minibatch. The sensitivity of the query is proportional to the maximum gradient norm $l$, and inversely proportional to the batch size $b$. By pertubing the gradient with a Gaussian noise of variance $\sigma^2 \frac{l^2}{b^2}$ the query is transformed into a Gaussian mechanism. By composing the Gaussian mechanisms we obtain the DPSGD variant, that enjoy $(\epsilon, \delta)$-DP guarantees.

**Proposition 1** (DP guarantees for SGD, adapted from Abadi et al. (2016).)**.** *Assume that the loss fulfills $\|\nabla_\theta \mathcal{L}(\hat{y}, y)\|_2 \le l$, and assume that the network is trained on a dataset of size $N$ with SGD algorithm for $T$ steps with noise scale $\mathcal{N}(\mathbf{0}, \sigma^2)$ such that:*

$$\sigma \ge \frac{16K\sqrt{T\log(2/\delta)\log(1.25T/\delta N)}}{N\epsilon}. \tag{12}$$

*Then the SGD training of the network is $(\epsilon, \delta)$-DP.*

## A.2 PRIVACY ACCOUNTING OF CLIPLESS DP-SGD

**The "global" strategy.** This strategy simply aggregates the individual sensitivities $\Delta_d$ of each layer to obtain the global sensitivity of the whole gradient vector $\Delta = \sqrt{\sum_d \Delta_d^2}$. The origin of the clipping-based version of this strategy can be traced back to McMahan et al. (2018). With noise variance $\sigma^2 \Delta^2$ we recover the accountant that comes with DP-SGD. It tends to overestimate the true sensitivity (in particular for deep networks), but its implementation is straightforward with existing tools. We detail in Algorithm 3 the global variant of our approach.

**The "per-layer" strategy.** Recall that we are able to characterize the sensitivity $\Delta_d$ of every layer of the network. Hence, we can apply a different noise to each of the gradients. We dissect the whole training procedure in Figure 6.

1. On each layer, we apply a Gaussian mechanism with noise variance $\sigma^2 \Delta_d^2$.

2. Their composition yields an other Gaussian mechanism with non isotropic noise.

3. The Gaussian mechanism benefits from privacy amplification via subsampling (Balle et al., 2018) thanks to the stochasticity in the selection of batches of size $b = pN$.

4. Finally an epoch is defined as the composition of $T = \frac{1}{p}$ sub-sampled mechanisms.

At same noise multiplier $\sigma$, "per-layer" strategy tends to produce a higher value of $\epsilon$ per epoch than the "global" strategy, but has the advantage over the latter to add smaller effective noise $\zeta$ to each weight. Different layers exhibit different maximum gradient bounds - and in turn this implies different sensitivities. This also suggests that different noise multipliers $\sigma_d$ can be used for each layer. This open extensions for future work.

For practical computations, we rely on the `autodp` library (Wang et al., 2019; Zhu & Wang, 2019; 2020) as it uses the Rényi Differential Privacy (RDP) adaptive composition theorem (Mironov, 2017; Mironov et al., 2019), that ensures tighter bounds than naive DP composition.

---

**Algorithm 3 Clipless DP-SGD** with **global** sensitivity accounting

---

**Input**: Feed-forward architecture $f(\cdot, \cdot) = f_{T+1}(\theta_{T+1}, \cdot) \circ f_T(\theta_T, \cdot) \circ \ldots \circ f_0(\theta_0, \cdot)$
**Input**: Initial weights $\theta_0$, learning rate scheduling $\eta_t$, noise multiplier $\sigma$.

1: **repeat**
2:     $\Delta_0 \ldots \Delta_{T+1} \leftarrow$ compute_gradient_bounds$(f, X)$.
3:     Sample a batch $\mathcal{B}_t = \{(x_1, y_1), (x_2, y_2), \ldots, (x_b, y_b)\}$.
4:     Compute the mean gradient of the batch for each layer $t$:

$$\tilde{g}_t := \frac{1}{b} \sum_{i=1}^{b} \nabla_{\theta_t} \mathcal{L}(\hat{y}_i, y_i)).$$

5:     For each layer $t$ of the model, get the theoretical bound of the gradient:

$$\forall 1 \leq i \leq b, \quad \|\nabla_{\theta_t} \mathcal{L}(\hat{y}_i, y_i)\|_2 \leq \Delta_t.$$

6:     Update Moment accountant state with **global** sensitivity $\Delta = \frac{2}{b} \sqrt{\sum_{t=1}^{T+1} \Delta_t^2}$.
7:     Add global noise $\zeta \sim \mathcal{N}(0, 2\sigma\Delta/b)$ to each weights and perform projected gradient step:

$$\theta_t \leftarrow \Pi(\theta_t - \eta(\tilde{g}_t + \zeta)).$$

8:     Report new $(\epsilon, \delta)$-DP guarantees with accountant.
9: **until** privacy budget $(\epsilon, \delta)$ has been reached.

---

## A.3 ROBUSTNESS CERTIFICATES OF LIPSCHITZ CONSTRAINED NETWORKS

The temperature parameter $\tau$ of softmax has a strong influence on the certificates. The models with the most robust decisions are not the ones with the best clean accuracy, which is compliant with the observations of Béthune et al. (2022).

| Softmax Temperature | Certifiable accuracy at radius $r/255$ | | | | | | Membership Inference Attacks | |
|---|---|---|---|---|---|---|---|---|
| | $r = 0$ | $r = 1$ | $r = 2$ | $r = 4$ | $r = 8$ | $r = 16$ | AUROC | Adv. |
| $\tau = 0.01$ | **47.4** | 5.9 | 0.1 | 0.0 | 0.0 | 0.0 | 59.8 | 23.9 |
| $\tau = 0.22$ | 44.4 | **39.1** | 34.2 | 24.9 | 11.5 | 0.8 | 50.1 | 15.0 |
| $\tau = 0.40$ | 41.6 | 38.4 | **35.6** | 29.6 | 19.1 | 5.3 | 50.0 | 11.3 |
| $\tau = 0.74$ | 38.4 | 36.4 | 34.7 | **30.9** | 24.1 | 13.0 | 51.9 | 20.5 |
| $\tau = 2.77$ | 33.3 | 32.2 | 31.2 | 29.3 | **25.9** | 18.8 | 52.5 | 21.3 |
| $\tau = 5.40$ | 32.5 | 31.4 | 30.4 | 28.8 | 25.5 | **19.7** | 59.7 | 23.6 |

Table 1: **Certifiable accuracy scores under $(20.0, 10^{-5})$-DP training on Cifar-10 dataset**. For each targeted robustness radius $r/255$, we optimize over softmax-crossentropy temperature $\tau$, and report the best value. We also monitor the best AUROC and the best attacker Advantage (Adv.) Yeom et al. (2018) achieved by two popular membership inference attacks: threshold attacks Song & Mittal (2021) and a logistic regression shadow model Shokri et al. (2017).

## A.4 BIAS OF CLIPLESS DP-SGD

Clipless DP-SGD *also* exhibits a bias, but this bias takes a different form than the bias induced by clipping in DP-SGD.

**The bias of the optimizer.** In DP-SGD (with clipping) the average gradient is biased by the elementwise clipping. Therefore, the clipping may slow down convergence or lead to sub-optimal solutions.

**The bias of the model in the space in Lipschitz networks.** This is a bias of the model, not of the optimizer. It has been shown that any classification task could be solved with a 1-Lipschitz classifier (Béthune et al., 2022), and in this sense, the bias induced by the space of 1-Lipschitz functions is not too severe. Better, this bias is precisely what allows to produce robustness certificates, see for example Yang et al. (2020).

**Finally, there is the implicit bias.** It is induced by a given architecture on the optimizer, which can have strong effects on effective generalization. For neural networks (Lipschitz or not), this implicit bias is not fully understood yet. But even on large Lipschitz models, it seems that the Lipschitz constraint biases the network toward better robustness radii but worse clean accuracy, as frequently observed in the relevant literature.

Therefore, these biases influence the learning process differently, and they constitute two distinct (not necessarily exclusive) approaches. We illustrate the difference between these paradigms in Figure 7.

## A.5 EFFICIENCY OF GRADIENT CLIPPING

Beyond the conventional implementation of gradient clipping that can be found in Opacus or Tf-privacy, recent developments proposed more efficient forms of clipping, or to get rid of the clipping introduced by Abadi et al. (2016) and to use renormalization instead.

In this regard, we can mention the works of Bu et al. (2022b) or Yang et al. (2022) that study alternative implementations of clipping based on renormalization, which eliminates the need to tune the clipping value, with convergence guarantees.

Other works study the alternative implementation of elementwise clipping to reduce the computational cost, like Bu et al. (2022a), Li et al. (2021) He et al. (2022), and Bu et al. (2023). Taking inspiration from Bu et al. (2023) we summarize each one in Tables 2 and 3.

## B LIPDP TUTORIAL

This section gives advice on how to start your DP training processes using the framework. Moreover, it provides insights into how input pre-processing, building networks with Residual connections and Loss-logits gradient clipping could help offer better utility for the same privacy budget.

| | Instantiating per-sample gradient | Storing every layer's gradient | Instantiating non-DP gradient | Number of back-propagations | Overhead independent of batch size |
|---|---|---|---|---|---|
| non-DP | ✗ | ✗ | ✓ | 1 | ✓ |
| TF-Privacy, like Abadi et al. (2016) | ✓ | ✗ | ✓ | B | ✗ |
| Opacus (Yousefpour et al., 2021) | ✓ | ✓ | ✓ | 1 | ✗ |
| FastGradClip (Lee & Kifer, 2021) | ✓ | ✗ | ✗ | 2 | ✗ |
| GhostClip (Li et al., 2021; Bu et al., 2022a) | ✗ | ✗ | ✓ | 1 | ✗ |
| Book-Keeping (Bu et al., 2023) | ✗ | ✗ | ✗ | 1 | ✗ |
| Clipless w/ Lipschitz (ours) | ✗ | ✗ | ✓ | 1 | ✓ |
| Clipless w/ GNP (ours) | ✗ | ✗ | ✓ | 1 | ✓ |

Table 2: **Comparison of DP-SGD against existing techniques of literature**. Clipless DP-SGD is the only technique whose time/memory overhead depends exclusively on weight matrices but not the batch size.

| | Time Complexity Overhead | Memory Overhead |
|---|---|---|
| non-DP | 0 | 0 |
| TF-Privacy, like Abadi et al. (2016) | $\mathcal{O}(BTpd)$ | 0 |
| Opacus (Yousefpour et al., 2021) | $\mathcal{O}(BTpd)$ | $\mathcal{O}(Bpd)$ |
| FastGradClip (Lee & Kifer, 2021) | $\mathcal{O}(BTpd)$ | $\mathcal{O}(Bpd)$ |
| GhostClip (Li et al., 2021; Bu et al., 2022a) | $\mathcal{O}(BTpd + BT^2)$ | $\mathcal{O}(BT^2)$ |
| Book-Keeping (Bu et al., 2023) | $\mathcal{O}(BTpd)$ | $\mathcal{O}(B\min(pd, T^2))$ |
| Clipless w/ Lipschitz (ours) | $\mathcal{O}(Upd)$ | 0 |
| Clipless w/ GNP (ours) | $\mathcal{O}(Upd + Vpd\min(p,d))$ | 0 |

Table 3: **Time and memory costs for each method** for feedforward networks. We assume weight matrices of shape $p \times d$, and a *physical* batch size $B$. For images, $T$ is height×width. $U$ is the number of iterations in Power Iteration algorithm, and $V$ the number of iterations in Björck projection. Typically $U, V < 15$ in practice. The overhead is taken relatively to non-DP training *without* clipping. This table is largely inspired from Bu et al. (2023).

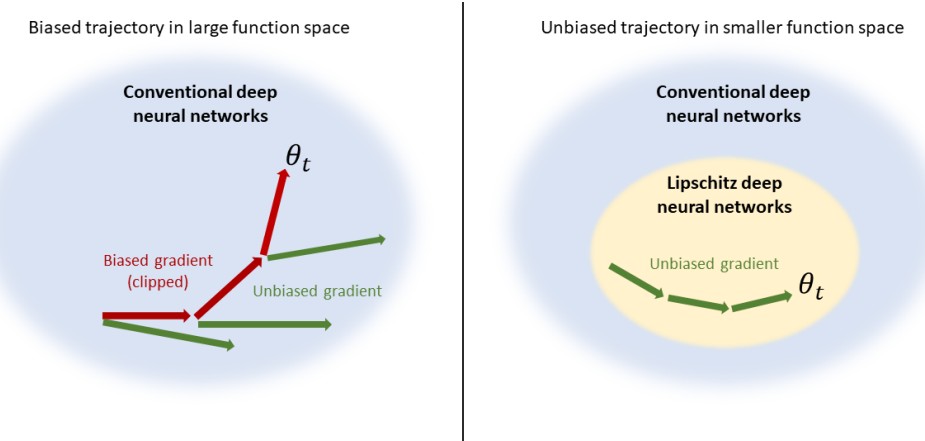

Figure 7: Comparison between the bias of DP-SGD *with* clipping, and Clipless DP-SGD. One is an instance of *optimizer bias*, and the other of *function space bias*.

As per def 3 a $l$-Lipschitz neural network of depth $d$ can be built by composing $\sqrt[d]{l}$ layers. In the rest of this section we will focus on 1-Lipschitz networks (rather than controlling $l$ we control the loss to obtain the same effects Béthune et al. (2022)). In order to do so the strategy consists in choosing only 1-Lipschitz activations, and to constrain the weights of parameterized layers such that it can only express 1-Lipschitz functions. For instance normalizing the weight matrix of a dense layer by its spectral norm yield a 1-Lipschitz layer (this, however cannot be applied trivially on convolution's kernel). In practice we used the layers available in the open-source library *deel-lip*. In practice, when building a Lipschitz network, the following block can be used:

- Dense layers: are available as *SpectralDense* or *QuickSpectralDense* which apply spectral normalization and Björck orthogonalization, making them GNP layers.
- *Relu* activations are replaced by *Groupsort2* activations, and such activations are GNP, preventing vanishing gradient.
- pooling layers are replaced with *ScaledL2NormPooling*, which is GNP since it performs the operation $x \mapsto (\|x_m\|_2)_m$ where $m$ is the multi-index of a patch of contiguous pixels.
- normalization layers like *BatchNorm* are not $K$-Lipschitz. We did not accounted these layers since they can induce a privacy leak (as they keep a rolling mean and variance). A 1-Lipschitz drop-in replacement is studied in C.2.3. The relevant literature also propose drop-in replacement for this layer with proper sensitivity accounting.

Originally, this library relied on differentiable re-parametrizations, since it yields higher accuracy outside DP training regime (clean training without noise). However, our framework does not account for the Lipschitz constant of the re-parametrization operator. This is why we provide *QuickSpectralDense* and *QuickSpectralConv2D* layers to enforce Lipschitz constraints, where the projection is enforced with a tensorflow constraint. Note that *SpectralDense* and *SpectralConv2D* can still be used with a *CondenseCallack* to enforce the projection, and bypass the back-propagation through the differentiable re-parametrization. However this last solution, while being closer to the original spirit of deel-lip, is also less efficient in speed benchmarks.

The Lipschitz constant of each layer is bounded by 1, since each weight matrix is divided by the largest singular value $\sigma_{\max}$. This singular value is computed with Power Iteration algorithm. Power Iteration computes the largest singular value by repeatedly computing a Rayleigh quotient asssociated to a vector $u$, and this vector eventually converges to the eigenvector $u_{\max}$ associated to the largest eigenvalue. These iterations can be expensive. However, since gradient steps are smalls, the weight matrices $W_t$ and $W_{t+1}$ remain close to each other after a gradient step. Hence their largest eigenvectors tend to be similar. Therefore, the eigenvector $u_{\max}$ can be memorized at the end of each train step, and re-used as high quality initialization at the next train step, in a "lazy" fashion. This speed-up makes the overall projection algorithm very efficient.

## B.2 GETTING STARTED

The framework we propose is built to allow DP training of neural networks in a fast and controlled approach. The tools we provide are the following :

1. **A pipeline** to efficiently load and pre-process the data of commonly used datasets like MNIST, FashionMNIST and CIFAR10.
2. **Configuration objects** to correctly account DP events we provide config objects to fill in that will
3. **Model objects** on the principle of Keras' model classes we offer both a `DP_Model` and a `DP_Sequential` class to streamline the training process.
4. **Layer objects** where we offer a readily available form of the principal layers used for DNNs. These layers are already Lipschitz constrained and possess class specific methods to access their Lipschitz constant.
5. **Loss functions**, identically, we offer DP loss functions that automatically compute their Lipschitz constant for correct DP enforcing.

We highlight below an example of a full training loop on Mnist with **lip-dp** library. Refer to the "examples" folder in the library for more detailed explanations in a jupyter notebook.

```python
dp_parameters = DPParameters(
    noisify_strategy="global",
    noise_multiplier=2.0,
    delta=1e-5,
)

epsilon_max = 3.0

input_upper_bound = 20.0
ds_train, ds_test, dataset_metadata = load_and_prepare_data(
    "mnist",
    batch_size=1000,
    drop_remainder=True,
    bound_fct=bound_clip_value(
        input_upper_bound
    ),
)

# construct DP_Sequential
model = DP_Sequential(
    layers=[
        layers.DP_BoundedInput(
            input_shape=dataset_metadata.input_shape, upper_bound=
                input_upper_bound
        ),
        layers.DP_QuickSpectralConv2D(
            filters=32,
            kernel_size=3,
            kernel_initializer="orthogonal",
            strides=1,
            use_bias=False,
        ),
        layers.DP_GroupSort(2),
        layers.DP_ScaledL2NormPooling2D(pool_size=2, strides=2),
        layers.DP_QuickSpectralConv2D(
            filters=64,
            kernel_size=3,
            kernel_initializer="orthogonal",
            strides=1,
            use_bias=False,
        ),
        layers.DP_GroupSort(2),
        layers.DP_ScaledL2NormPooling2D(pool_size=2, strides=2),

        layers.DP_Flatten(),

        layers.DP_QuickSpectralDense(512),
        layers.DP_GroupSort(2),
        layers.DP_QuickSpectralDense(dataset_metadata.nb_classes),
    ],
    dp_parameters=dp_parameters,
    dataset_metadata=dataset_metadata,
)

model.compile(
    loss=losses.DP_TauCategoricalCrossentropy(18.0),
    optimizer=tf.keras.optimizers.SGD(learning_rate=2e-4, momentum=0.9),
    metrics=["accuracy"],
)
model.summary()

num_epochs = get_max_epochs(epsilon_max, model)
```

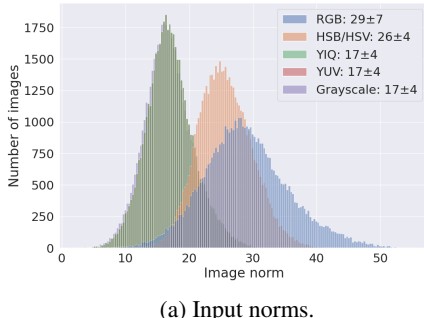
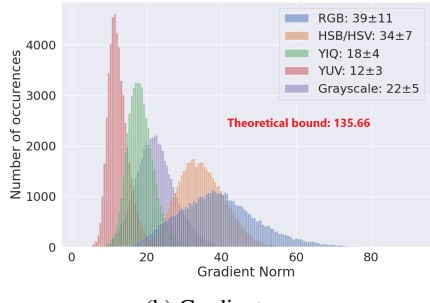

(a) Input norms.          (b) Gradient norms.

Figure 8: **Histogram of norms for different image space on CIFAR-10 images**. We see that for GNP networks the distribution of dataset norms have a strong influence on the norm of the individual parameterwise gradient norms of missclassified examples. The global gradient bound is equal to 135.66 on this architecture: less than two times the maximum empirical bound of 73.

```
hist = model.fit(
    ds_train,
    epochs=num_epochs,
    validation_data=ds_test,
    callbacks=[
        # accounting is done thanks to a callback
        DP_Accountant(log_fn="logging"),
    ],
)
```

## B.3    IMAGE SPACES AND INPUT CLIPPING

Input preprocessing can be done in a completely dataset agnostic way (i.e without privacy leakages) and may yield positive results on the models utility. We explore here the choice of the color space, and the norm clipping of the input.

### B.3.1    COLOR SPACE REPRESENTATIONS

The color space of images can yield very different gradient norms during the training process. Therefore, we can take advantage of this to train our DP models more efficiently. Empirically, Figures 8a and 8b show that some color spaces yield narrower image norm distributions that happen to be more advantageous to maximise the mean gradient norm to noise ratio across all samples during the DP training process of GNP networks.

### B.3.2    INPUT CLIPPING

A clever way to narrow down the distribution of the dataset's norms would be to clip the norms of the input of the model. This may result in improved utility since a narrower distribution of input norms might maximise the mean gradient norm to noise ratio for misclassified examples. Also, we advocate for the use of GNP networks as their gradients usually turn out to be closer to the upper bound we are able to compute for the gradient. See Figure 9a and 9b

## B.4    PRACTICAL IMPLEMENTATION OF RESIDUAL CONNECTIONS

The implementation of skip connections is made relatively straightforward in our framework by the `make_residuals` method. This function splits the input path in two, and wraps the layers inside the residual connections, as illustrated in figure 10.

```
from lipdp.layers import make_residuals
```

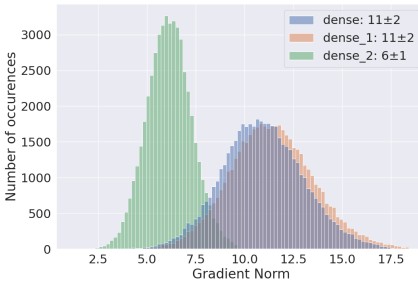
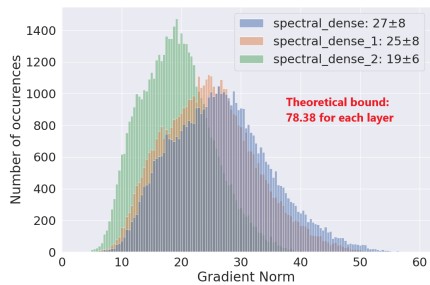

(a) Conventional network.

(b) Gradient Norm Preserving network.

Figure 9: **Gradient norms of fully-connected networks on CIFAR-10**. We see that GNP networks exhibit a qualitatively different profile of gradient norms with respect to parameters, sticking closer to the upper bound we are able to compute for the gradient norm. The GNP network does not use biases, therefore all layers hare the same bound of $78.38$ for their gradient norm. Not too far away from the maximum empirical bound of $56$ at initialization.

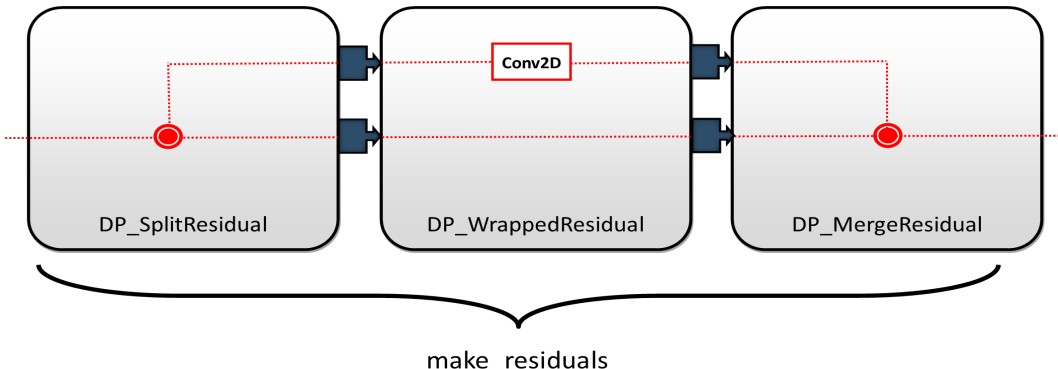

make_residuals

Figure 10: Implementation of a skip connection in the **lip-dp** framework. The meta block **DP_WrappedResidual** handle the forward propagation and the backward propagation of pairs of bounds (one for each computation path) by leveraging the forward and the backward of sub-blocks. **DP_SplitResidual** handle the creation of a tuple of input bounds at the forward, and collapse the tuple of gradient bounds into a scalar at the backward, while **DP_MergeResidual** does the opposite. All those operations are wrapped under the convenience function **make_residuals**.

```
## Manual implementation of residual connection:
layers = [DP_SplitResidual(),
          DP_WrappedResidual(DP_QuickSpectralConv2D(16, (3, 3)),
          DP_WrappedResidual(DP_GroupSort(2)),
          DP_MergeResidual('1-lip-add')]
## Or equivalently, with helper function:
layers = make_residuals([
    DP_QuickSpectralConv2D(16, (3, 3),
    DP_GroupSort(2)
 ])
```

By using this implementation, the sensitivity of the gradient computation and the input bounds to each layer are correctly computed when the model's path is split. This allows for fairly easy implementations of models like the MLP-Mixer and ResNets. Since convolutional models may suffer from gradient vanishing and that dense based models are relatively restrictive in terms of architecture, implementing skip connections could be a useful feature for our framework.

### B.5 Loss-logits gradient clipping

The advantage of the traditional DP-SGD approach is that through hyperparameter optimization on the global gradient clipping constant, we indirectly optimize the mean signal to noise ratio on missclassified examples. However, this clipping constant is not really explainable and rather just an empirical result of optimization on a given architecture and loss function.

Importantly, our framework is compatible with a more efficient and explainable form of clipping. As more and more examples are correctly classified, the magnitude of $\|\nabla_f \mathcal{L}\|$ tend to diminish over the course of training, quickly falling below the upper bound and diminishing the gradient norm to noise standard deviation ratio. Gradient clipping strategy is still available in our framework. But instead of clipping the parameter gradient $\nabla_\theta \mathcal{L}$, any intermediate gradient $\nabla_{f_d} \mathcal{L}$ can be clipped during backpropagation. Indeed, by introducing a "*clipping layer*" that behave like identity function at the forward pass, and clip the gradient during the backward pass, we are able to clip the gradient of the loss on the final logits for a minimal cost.

Indeed, in this case the size of the clipped vector is the output dimension $|\hat{y}|$, which is small for a lot of practical regression and classification tasks. For example in CIFAR-10 the output vector is of length 10. This scales better with the batch size than the weight matrices that are typically of sizes $64 \times 64$ or $128 \times 128$. More precisely, in vanilla DP-SGD the per-weight gradient clipping is applied on the batched gradient $\nabla_{W_d} \mathcal{L}$ which is a tensor of size $b \times h^2$ for matrix weight $W_d \in \mathbb{R}^{h \times h}$. In privacy it is not unusual to have $b$ and $h$ in the $[10^2, 10^4]$ range. Clipping this vector can cause memory issues or slowdowns Lee & Kifer (2021). However $\nabla_f \mathcal{L}$ is of size $b \times h$ which is significantly cheaper to clip. Note that the resulting "gradient" vector is not a true gradient anymore in a strictly mathematical sense, but rather a "descent direction" Boyd & Vandenberghe (2004).

Furthermore, if the gradient clipping layer is inserted on the tail of the network (between the logits and the loss) we can characterize its effects on the training, in particular for classification tasks with binary cross-entropy loss.

We denote by $\mathcal{L}(\text{Clip}_\nabla^C(\hat{y}))$ the loss wrapped under a **DP_ClipGradient** layer that behaves like identity $x \mapsto x$ at the forward, and clips the gradient norm to $C > 0$ in the backward pass:

$$\nabla_{\hat{y}} \left( \mathcal{L}(\text{Clip}_\nabla^C(\hat{y})) \right) := \min \left( 1, \frac{C}{\|\nabla_{\hat{y}} \mathcal{L}(\hat{y})\|} \right) \nabla_{\hat{y}} \mathcal{L}(\hat{y}) \tag{13}$$

$$= \min \left( \|\nabla_{\hat{y}} \mathcal{L}(\hat{y})\|, C \right) \frac{\nabla_{\hat{y}} \mathcal{L}(\hat{y})}{\|\nabla_{\hat{y}} \mathcal{L}(\hat{y})\|}. \tag{14}$$

We denote by $\overrightarrow{g(\hat{y})}$ the unit norm vector $\frac{\nabla_{\hat{y}} \mathcal{L}(\hat{y})}{\|\nabla_{\hat{y}} \mathcal{L}(\hat{y})\|}$. Then:

$$\nabla_{\hat{y}} \left( \mathcal{L}(\text{Clip}_\nabla^C(\hat{y})) \right) = \min \left( \|\nabla_{\hat{y}} \mathcal{L}(\hat{y})\|, C \right) \overrightarrow{g(\hat{y})}.$$

**Proposition 2** (Clipped binary cross-entropy loss is the Wasserstein dual loss)**.** *Let $\mathcal{L}_{BCE}(\hat{y}, y) = -\log(\sigma(\hat{y}y))$ be the binary cross-entropy loss, with $\sigma(\hat{y}y) = \frac{1}{1 + \exp(-\hat{y}y)}$ the sigmoid activation, assuming discrete labels $y \in \{-1, +1\}$. Assume examples are sampled from the dataset $\mathcal{D}$. Let $\hat{y}(\theta, x) = f(\theta, x)$ be the predictions at input $x \in \mathcal{D}$. Then for every $C > 0$ sufficiently small, a gradient descent step with the clipped gradient $\nabla_\theta \mathbb{E}_\mathcal{D}[\mathcal{L}(\text{Clip}_\nabla^C(\hat{y}, y))]$ is identical to the gradient ascent step obtained from Kantorovich-Rubinstein loss $\mathcal{L}_{KR}(\hat{y}, y) = \hat{y}y$.*

*Proof.* In the following we use the short notation $\mathcal{L}$ in place of $\mathcal{L}_{BCE}$. Assume that examples with labels $+1$ (resp. $-1$) are sampled from distribution $P$ (resp. $Q$). By definition:

$$\nabla_\theta \mathbb{E}_\mathcal{D}[\mathcal{L}(\text{Clip}_\nabla^C(\hat{y}, y))] = \nabla_\theta \left( \mathbb{E}_{x \sim P}[\mathcal{L}(\text{Clip}_\nabla^C(f(\theta, x), +1))] + \mathbb{E}_{x \sim Q}[\mathcal{L}(\text{Clip}_\nabla^C(f(\theta, x), -1))] \right).$$

Observe that the output of the network is a single scalar, hence $\overrightarrow{g(\hat{y})} \in \{-1, +1\}$. We apply the chainrule $\nabla_\theta \mathcal{L} = \nabla_{\hat{y}} \mathcal{L} \frac{\partial \hat{y}}{\partial \theta} = \overrightarrow{g(\hat{y})} \min(\|\nabla_{\hat{y}} \mathcal{L}(\hat{y})\|, C) \nabla_\theta f(\theta, x)$. We note $R(\hat{y}, C) := \min(\|\nabla_{\hat{y}} \mathcal{L}(\hat{y})\|, C) > 0$ and we obtain:

$$\nabla_\theta \mathbb{E}_\mathcal{D}[\mathcal{L}(\text{Clip}_\nabla^C(\hat{y}, y))] = \nabla_\theta \left( \mathbb{E}_{x \sim P}[\overrightarrow{g(\hat{y})} R(\hat{y}, C) \nabla_\theta f(\theta, x)] + \mathbb{E}_{x \sim Q}[\overrightarrow{g(\hat{y})} R(\hat{y}, C) \nabla_\theta f(\theta, x)] \right). \tag{15}$$

Observe that the value of $\overrightarrow{g(\hat{y})}$ can actually be deduced from the label $y$, which gives:

$$\nabla_\theta \mathbb{E}_\mathcal{D}[\mathcal{L}(\hat{y}, y)] = -\mathbb{E}_{x \sim P}[R(\hat{y}, C)\nabla_\theta f(\theta, x)] + \mathbb{E}_{x \sim Q}[R(\hat{y}, C)\nabla_\theta f(\theta, x)]. \qquad (16)$$

Observe that the function $x \mapsto \nabla_{\hat{y}}\mathcal{L}(\hat{y})$ is piecewise-continuous when the loss $\hat{y} \mapsto \mathcal{L}(\hat{y}, y)$ is piecewise continuous. Observe that $|\nabla_{\hat{y}}\mathcal{L}(\hat{y})|$ is non zero, since the loss $\mathcal{L}$ does not achieve its minimum over the open set $(-\infty, +\infty)$, since $\sigma(\hat{y}) \in (0, 1)$. Assuming that the data $x \in \mathcal{D}$ lives in a compact space (or equivalently that $P$ and $Q$ have compact support), since $x \mapsto |\nabla_{\hat{y}}\mathcal{L}(\hat{y})|$ is piecewise continuous (with finite number of pieces for finite neural networks) it attains its minimum $C' > 0$. Choosing any $C < C'$ implies that $R(\hat{y}, C) = C$, which yields:

$$\nabla_\theta \mathbb{E}_\mathcal{D}[\mathcal{L}(\text{Clip}_\nabla^C(\hat{y}, y))] = -\mathbb{E}_{x \sim P}[C\nabla_\theta f(\theta, x)] + \mathbb{E}_{x \sim Q}[C\nabla_\theta f(\theta, x)]$$
$$= -C\left(\mathbb{E}_{x \sim P}[\nabla_\theta f(\theta, x)] - \mathbb{E}_{x \sim Q}[\nabla_\theta f(\theta, x)]\right).$$

This corresponds to a gradient *ascent* step of length $C$ on the Kantorovich-Rubinstein (KR) objective $\mathcal{L}_{KR}(\hat{y}, y) = \hat{y}y$. This loss is named after the Kantorovich Rubinstein duality that arises in optimal transport, than states that the Wasserstein-1 distance is a supremum over 1-Lipschitz functions:

$$\mathcal{W}_1(P, Q) := \sup_{f \in 1\text{-Lip}(\mathcal{D}, \mathbb{R})} \mathbb{E}_{x \sim P}[f(x)] - \mathbb{E}_{x \sim Q}[f(x)]. \qquad (17)$$

Hence, with clipping $C$ small enough the gradient steps are actually identical to the ones performed during the estimation of Wasserstein-1 distance. $\qquad\square$

The adaptive clipping of Andrew et al. (2021) takes the form of a Gaussian mechanism so the implementation of the RDP accountant is straightforward.

In future works, other multi-class losses can be studied through the lens of per-example clipping. Our framework permits the use of gradient clipping while at the same time facilitating the theoretical anaysis.

### B.5.1 POSSIBLE IMPROVEMENTS

Our framework is compatible with possible improvements in methods of data pre-processing. For instance some works suggest that feature engineering is the key to achieve correct utility/privacy trade-off Tramer & Boneh (2021) some other work rely on heavily over-parametrized networks, coupled with batch size De et al. (2022). While we focused on providing competitive and reproducible baselines (involving minimal pre-processing and affordable compute budget) our work is fully compatible with those improvements. Secondly the field of GNP networks (also called orthogonal networks) is still an active field, and new methods to build better GNP networks will improve the efficiency of our framework ( for instance orthogonal convolutions Achour et al. (2022),Li et al. (2019b),Li et al. (2019a),Trockman & Kolter (2021),Singla & Feizi (2021b) are still an active topic). Finally some optimizations specific to our framework can also be developed: a scheduling of loss-logits clipping might allow for better utility scores by following the declining value of the gradient of the loss, therefore allowing for a better mean signal to noise ratio across a diminishing number of miss-classified examples. Observe that our framework is also compatible with other ideas, such as the progressing freezing of deeper layers proposed by Tang & Lécuyer (2022).

## C COMPUTING SENSITIVITY BOUNDS

### C.1 LIPSCHITZ CONSTANTS OF COMMON LOSS FUNCTIONS

This section contains the proofs related to the content of Table 4. Our framework wraps over some losses found in deel-lip library, that are wrapped by our framework to provide Lipschitz constant automatically during backpropagation for bounds.

**Multiclass Hinge** This loss, with min margin $m$ is computed in the following manner for a one-hot encoded ground truth vector $y$ and a logit prediction $\hat{y}$ :

$$\mathcal{L}_{MH}(\hat{y}, y) = \{\max(0, \frac{m}{2} - \hat{y}_1 . y_1), ..., \max(0, \frac{m}{2} - \hat{y}_k . y_k)\}.$$

And $\|\frac{\partial}{\partial y}\mathcal{L}_{MH}(\hat{y}, y)\|_2 \leq \|\hat{y}\|_2$. Therefore $L_H = 1$.

| Loss | Hyper-parameters | $\mathcal{L}(\hat{y}, y)$ | Lipschitz bound $L$ |
|---|---|---|---|
| Softmax Cross-entropy | temperature $\tau > 0$ | $y^T \log \text{softmax}(\hat{y}/\tau)$ | $\sqrt{2}/\tau$ |
| Cosine Similarity | bound $X_{\min} > 0$ | $\frac{y^T \hat{y}}{\max(\|\hat{y}\|, X_{\min})}$ | $1/X_{\min}$ |
| Multiclass Hinge | margin $m > 0$ | $\{\max(0, \frac{m}{2} - \hat{y}_i . y_i)\}_{1 \le i \le K}$ | $1$ |
| Kantorovich-Rubenstein | N/A | $\{\hat{y}, y\}$ | $1$ |
| Hinge Kantorovich-Rubenstein | margin $m > 0$
regularization $\alpha > 0$ | $\alpha . \mathcal{L}_{MH}(\hat{y}, y) + \mathcal{L}_{MKR}(\hat{y}, y)$ | $1 + \alpha$ |

Table 4: Lipschitz constant of common supervised classification losses used for the training of Lipschitz neural networks with $k$ classes. Proofs in Section C.1.

**Multiclass Kantorovich Rubenstein** This loss, is computed in a one-versus all manner, for a one-hot encoded ground truth vector $y$ and a logit prediction $\hat{y}$ :

$$\mathcal{L}_{MKR}(\hat{y}, y) = \{\hat{y}_1 - y_1, \ldots, \hat{y}_k - y_k)\}.$$

Therefore, by differentiating, we also get $L_{KR} = 1$.

**Multiclass Hinge - Kantorovitch Rubenstein** This loss, is computed in the following manner for a one-hot encoded ground truth vector $y$ and a logit prediction $\hat{y}$ :

$$\mathcal{L}_{MHKR}(\hat{y}, y) = \alpha \mathcal{L}_{MH}(\hat{y}, y) + \mathcal{L}_{MKR}(\hat{y}, y).$$

By linearity we get $L_{HKR} = \alpha + 1$.

**Cosine Similarity** Cosine Similarity is defined in the following manner element-wise :

$$\mathcal{L}_{CS}(\hat{y}, y) = \frac{\hat{y}^T y}{\|\hat{y}\|_2 \|y\|_2}.$$

And $y$ is one-hot encoded, therefore $\mathcal{L}_{CS}(\hat{y}, y) = \frac{\hat{y}_i}{\|\hat{y}\|_2}$. Therefore, the Lipschitz constant of this loss is dependant on the minimum value of $\hat{y}$. A reasonable assumption would be $\forall x \in \mathcal{D}$ : $X_{min} \le \|x\|_2 \le X_{max}$. Furthermore, if the networks are Norm Preserving with factor K, we ensure that:
$$KX_{min} \le \|\hat{y}\|_2 \le KX_{max}.$$

Which yields: $L_{CS} = \frac{1}{KX_{min}}$. The issue is that the exact value of $K$ is never known in advance since Lipschitz networks are rarely purely Norm Preserving in practice due to various effects (lack of tightness in convolutions, or rectangular matrices that can not be perfectly orthogonal).

Realistically, we propose the following loss function in replacement:

$$\mathcal{L}_{K-CS}(\hat{y}, y) = \frac{\hat{y}_i}{\max(KX_{min}, \|\hat{y}\|_2)}.$$

Where $K$ is an input given by the user, therefore enforcing $L_{K-CS} = \frac{1}{KX_{min}}$.

**Categorical Cross-entropy from logits** The logits are mapped into the probability simplex with the *Softmax* function $\mathbb{R}^K \to (0, 1)^K$. We also introduce a temperature parameter $\tau > 0$, which hold signfiance importance in the accuracy/robustness tradeoff for Lipschitz networks as observed by Béthune et al. (2022). We assume the labels are discrete, or one-hot encoded: we do not cover the case of label smoothing.

$$S_j = \frac{\exp(\tau \hat{y}_j)}{\sum_i \exp(\tau \hat{y}_i)}. \tag{18}$$

| Layer | Hyper parameters | $\|\frac{\partial f_t(\theta_t, x)}{\partial \theta_t}\|_2$ |
|---|---|---|
| 1-Lipschitz dense | none | 1 |
| Convolution | window $s$ | $\sqrt{s}$ |
| RKO convolution | window $s$ image size $H \times W$ | $\sqrt{1/((1 - \frac{(h-1)}{2H})(1 - \frac{(w-1)}{2W}))}$ |

Table 5: Lipschitz constant with respect to parameters in common Lipschitz layers. We report only the multiplicative factor that appears in front of the input norm $\|x\|_2$.

| Layer | Hyper parameters | $\|\frac{\partial f_t(\theta_t, x)}{\partial x}\|_2$ |
|---|---|---|
| Add bias | none | 1 |
| 1-Lipschitz dense | none | 1 |
| RKO convolution | none | 1 |
| Layer centering | none | 1 |
| Residual block | none | 2 |
| ReLU, GroupSort softplus, sigmoid, tanh | none | 1 |

Table 6: Lipschitz constant with respect to intermediate activations.

We denote the prediction associated to the true label $j^+$ as $S_{j^+}$. The loss is written as:

$$\mathcal{L}(\hat{y}) = -\log(S_{j^+}). \tag{19}$$

Its gradient with respect to the logits is:

$$\nabla_{\hat{y}}\mathcal{L} = \begin{cases} \tau(S_{j^+} - 1) & \text{if } j = j^+, \\ \tau S_j & \text{otherwise} \end{cases} \tag{20}$$

The temperature factor $\tau$ is a multiplication factor than can be included in the loss itself, by using $\frac{1}{\tau}\mathcal{L}$ instead of $\mathcal{L}$. This formulation has the advantage of facilitating the tuning of the learning rate: this is the default implementation found in deel-lip library. The gradient can be written in vectorized form:

$$\nabla_{\hat{y}}\mathcal{L} = S - 1_{\{j=j^+\}}.$$

By definition of Softmax we have $\sum_{j \neq j^+} S_j^2 \leq 1$. Now, observe that $S_j \in (0, 1)$, and as a consequence $(S_{j^+} - 1)^2 \leq 1$. Therefore $\|\nabla_{\hat{y}}\mathcal{L}\|_2^2 = \sum_{j \neq j^+} S_j^2 + (S_{j^+} - 1)^2 \leq 2$. Finally $\|\nabla_{\hat{y}}\mathcal{L}\|_2 = \sqrt{2}$ and $L_{CCE} = \sqrt{2}$.

## C.2 LAYER BOUNDS

The Lipschitz constant (with respect to input) of each layer of interest is summarized in table 6, while the Lipschitz constant with respect to parameters is given in table 5.

### C.2.1 DENSE LAYERS

Below, we illustrate the basic properties of Lipschitz constraints and their consequences for gradient bounds computations. While for dense layers the proof is straightforward, the main ideas can be re-used for all linear operations which includes the convolutions and the layer centering.

**Property 2. Gradients for dense Lipschitz networks.** *Let $x \in \mathbb{R}^C$ be a data-point in space of dimensions $C \in \mathbb{N}$. Let $W \in \mathbb{R}^{C \times F}$ be the weights of a dense layer with $F$ features outputs. We bound the spectral norm of the Jacobian as*

$$\left\| \frac{\partial(W^T x)}{\partial W} \right\|_2 \leq \|x\|_2. \tag{21}$$

*Proof.* Since $W \mapsto W^T x$ is a linear operator, its Lipschitz constant is exactly the spectral radius:

$$\frac{\|W^T x - W'^T x\|_2}{\|W - W'\|_2} = \frac{\|(W - W')^T x\|_2}{\|W - W'\|_2} \leq \frac{\|W - W'\|_2 \|x\|_2}{\|W - W'\|_2} = \|x\|_2.$$

Finally, observe that the linear operation $x \mapsto W^T x$ is differentiable, hence the spectral norm of its Jacobian is equal to its Lipschitz constant with respect to $l_2$ norm. $\square$

### C.2.2 CONVOLUTIONS

**Property 3. Gradients for convolutional Lipschitz networks.** *Let $x \in \mathbb{R}^{S \times C}$ be an data-point with channels $C \in \mathbb{N}$ and spatial dimensions $S \in \mathbb{N}$. In the case of a time serie $S$ is the length of the sequence, for an image $S = HW$ is the number of pixels, and for a video $S = HWN$ is the number of pixels times the number of frames. Let $\Psi \in \mathbb{R}^{s \times C \times F}$ be the weights of a convolution with:*

- *window size $s \in \mathbb{N}$ (e.g $s = hw$ in 2D or $s = hwn$ in 3D),*

- *with $C$ input channels,*

- *with $F \in \mathbb{N}$ output channels.*

- *we don't assume anything about the value of strides. Our bound is typically tighter for strides=1, and looser for larger strides.*

*We denote the convolution operation as $(\Psi * \cdot) : \mathbb{R}^{S \times C} \to \mathbb{R}^{S \times F}$ with either zero padding, either circular padding, such that the spatial dimensions are preserved. Then the Jacobian of convolution operation with respect to parameters is bounded:*

$$\|\frac{\partial(\Psi * x)}{\partial \Psi}\|_2 \leq \sqrt{s} \|x\|_2. \tag{22}$$

*Proof.* Let $y = \Psi * x \in \mathbb{R}^{S \times F}$ be the output of the convolution operator. Note that $y$ can be uniquely decomposed as sum of output feature maps $y = \sum_{f=1}^{F} y^f$ where $y^f \in \mathbb{R}^{S \times F}$ is defined as:

$$\begin{cases} (y^f)_{if} = y_{if} & \text{for all } 1 \leq i \leq S, \\ (y^f)_{ij} = 0 & \text{if } j \neq f. \end{cases}$$

Observe that $(y^f)^T y^{f'} = 0$ whenever $f \neq f'$. As a consequence Pythagorean theorem yields $\|y\|_2^2 = \sum_{f=1}^{F} \|y^f\|_2^2$. Similarly we can decompose each output feature map as a sum of pixels $y^f = \sum_{p=1}^{S} y^{pf}$. where $y^{pf} \in \mathbb{R}^{S \times F}$ fulfill:

$$\begin{cases} (y^{pf})_{ij} = 0 & \text{if } i \neq p, j \neq f, \\ (y^{pf})_{pf} = y_{pf} & \text{otherwise.} \end{cases}$$

Once again Pythagorean theorem yields $\|y^f\|_2^2 = \sum_{p=1}^{S} \|y^{pf}\|_2^2$. It remains to bound $y^{pf}$ appropriately. Observe that by definition:

$$y^{pf} = (\Psi * x)_{pf} = (\Psi^f)^T x^p[s].$$

where $\Psi^f \in \mathbb{R}^{s \times C}$ is a slice of $\Psi$ corresponding to output feature map $f$, and $x^p[s] \in \mathbb{R}^{s \times C}$ denotes the patch of size $s$ centered around input element $p$. For example, in the case of images with $s = 3 \times 3$, $p$ are the coordinates of a pixel, and $x^p[s]$ are the input feature maps of $3 \times 3$ pixels around it. We apply Cauchy-Schwartz:

$$\|y^{pf}\|_2^2 \leq \|\Psi^f\|_2^2 \times \|x^p[s]\|_2^2.$$

By summing over pixels we obtain:

$$\|y^f\|_2^2 \le \|\Psi^f\|_2^2 \sum_{p=1}^{S} \|x^p[s]\|_2^2, \tag{23}$$

$$\implies \|y\|_2^2 \le \left(\sum_{f=1}^{F} \|\Psi^f\|_2^2\right)\left(\sum_{p=1}^{S} \|x^p[s]\|_2^2\right), \tag{24}$$

$$\implies \|y\|_2^2 \le \|\Psi\|_2^2 \times \left(\sum_{p=1}^{S} \|x^p[s]\|_2^2\right). \tag{25}$$

The quantity of interest is $\sum_{p=1}^{S} \|x^p[s]\|_2^2$ whose squared norm is the squared norm of all the patches used in the computation. With zero or circular padding, the norm of the patches cannot exceed those of input image. Note that each pixel belongs to atmost $s$ patches, and even exactly $s$ patches when circular padding is used:

$$\sum_{p=1}^{S} \|x^p[s]\|_2^2 \le s \sum_{p=1}^{S} \|x_p\|_2^2 = s\|x\|_2^2.$$

Note that when strides>1 the leading multiplicative constant is typically smaller than $s$, so this analysis can be improved in future work to take into account strided convolutions. Since $\Psi$ is a linear operator, its Lipschitz constant is exactly its spectral radius:

$$\frac{\|(\Psi * x) - (\Psi' * x)\|_2}{\|\Psi - \Psi'\|_2} = \frac{\|(\Psi - \Psi') * x\|_2}{\|\Psi - \Psi'\|_2} \le \frac{\sqrt{s}\|\Psi - \Psi'\|_2\|x\|_2}{\|\Psi - \Psi'\|_2} = \sqrt{s}\|x\|_2.$$

Finally, observe that the convolution operation $\Psi * x$ is differentiable, hence the spectral norm of its Jacobian is equal to its Lipschitz constant with respect to $l_2$ norm:

$$\|\frac{\partial(\Psi * x)}{\partial \Psi}\|_2 \le \sqrt{s}\|x\|_2.$$

$\square$

An important case of interest are the convolutions based on Reshaped Kernel Orthogonalization (RKO) method introduced by Li et al. (2019a). The kernel $\Psi$ is reshaped into 2D matrix of dimensions $(sC \times F)$ and this matrix is orthogonalised. This is not sufficient to ensure that the operation $x \mapsto \Psi * x$ is orthogonal - however it is 1-Lipschitz and only *approximately* orthogonal under suitable re-scaling by $\mathcal{N} > 0$.

**Corollary 1** (Loss gradient for RKO convolutions.). *For RKO methods in 2D used in Serrurier et al. (2021), the convolution kernel is given by $\Phi = \mathcal{N}\Psi$ where $\Psi$ is an orthogonal matrix (under RKO) and $\mathcal{N} > 0$ a factor ensuring that $x \mapsto \Phi * x$ is a 1-Lipschitz operation. Then, for RKO convolutions without strides we have:*

$$\|\frac{\partial(\Psi * x)}{\partial \Psi}\|_2 \le \sqrt{\frac{1}{(1 - \frac{(h-1)}{2H})(1 - \frac{(w-1)}{2W})}}\|x\|_2. \tag{26}$$

*where $(H, W)$ are image dimensions and $(h, w)$ the window dimensions. For large images with small receptive field (as it is often the case), the Taylor expansion in $h \ll H$ and $w \ll W$ yields a factor of magnitude $1 + \frac{(h-1)}{4H} + \frac{(w-1)}{4W} + \mathcal{O}(\frac{(w-1)(h-1)}{8HW}) \approx 1$.*

### C.2.3  LAYER NORMALIZATIONS

**Property 4. Bounded loss gradient for layer centering.** *Layer centering is defined as $f(x) = x - (\frac{1}{n}\sum_{i=1}^{n} x_i)\mathbf{1}$ where $\mathbf{1}$ is a vector full of ones, and acts as a "centering" operation along some channels (or all channels). Then the singular values of this linear operation are:*

$$\sigma_1 = 0, \quad and \quad \sigma_2 = \sigma_3 = \ldots = \sigma_n = 1. \tag{27}$$

*In particular $\|\frac{\partial f}{\partial x}\|_2 \le 1$.*

*Proof.* It is clear that layer normalization is an affine layer. Hence the spectral norm of its Jacobian coincides with its Lipschitz constant with respect to the input, which itself coincides with the spectral norm of $f$. The matrice $M$ associated to $f$ is symmetric and diagonally dominant since $|\frac{n-1}{n}| \geq \sum_{i=1}^{n-1} |\frac{-1}{n}|$. It follows that $M$ is semi-definite positive. In particular all its eigenvalues $\lambda_1 \leq \ldots \leq \lambda_n$ are non negative. Furthermore they coincide with its singular values: $\sigma_i = \lambda_i$. Observe that for all $r \in \mathbb{R}$ we have $f(r\mathbf{1}) = \mathbf{0}$, i.e the operation is null on constant vectors. Hence $\lambda_1 = 0$. Consider the matrix $M - I$: its kernel is the eigenspace associated to eigenvalue 1. But the matrix $M - I = \frac{-1}{n}\mathbf{1}\mathbf{1}^T$ is a rank-one matrix. Hence its kernel is of dimension $n - 1$, from which it follows that $\lambda_2 = \ldots = \lambda_n = \sigma_2 \ldots = \sigma_n = 1$. □

### C.2.4 MLP Mixer architecture

The MLP-mixer architecture introduced in Tolstikhin et al. (2021) consists of operations named *Token mixing* and *Channel mixing* respectively. For token mixing, the input feature is split in disjoint patches on which the same linear opration is applied. It corresponds to a convolution with a stride equal to the kernel size. convolutions on a reshaped input, where patches of pixels are "collapsed" in channel dimensions. Since the same linear transformation is applied on each patch, this can be interpreted as a block diagonal matrix whose diagonal consists of $W$ repeated multiple times. More formally the output of Token mixing takes the form of $f(x) := [W^T x_1, W^T x_2, \ldots W^T x_n]$ where $x = [x_1, x_2, \ldots, x_n]$ is the input, and the $x_i$'s are the patches (composed of multiple pixels). Note that $\|f(x)\|_2^2 \leq \sum_{i=1}^{n} \|W\|_2^2 \|x_i\|_2^2 = \|W\|_2^2 \sum_{i=1}^{n} \|x_i\|_2^2 = \|W\|_2^2 \|x\|_2^2$. If $\|W\|_2 = 1$ then the layer is 1-Lipschitz - it is even norm preserving. Same reasoning apply for Channel mixing. Therefore the MLP_Mixer architecture is 1-Lipchitz and the weight sensitivity is proportional to $\|x\|$.

**Lipschitz MLP mixer:** We adapted the original architecture in order to have an efficient 1-Lipschitz version with the following changes:

- Relu activations were replaced with GroupSort, allowing a better gradient norm preservation,
- Dense layers were replaced with their GNP equivalent,
- Skip connections are available (adding a 0.5 factor to the output in order to ensure 1-lipschitz condition) but architecture perform as well without these.

Finally the architecture parameters were selected as following:

1. The number of layer is reduced to a small value (between 1 and 4) to take advantage of the theoretical sensitivity bound.

2. The patch size and hidden dimension are selected to achieve a sufficiently expressive network (a patch size between 2 and 4 achieves sufficient accuracy without over-fitting, and a hidden dimension of 128-512 unlocks allows batch size).

3. The channel dim and token dimensions are chosen such that weight matrices are square matrices (exact gradient norm preservation property requires square matrices).

### C.3 Tightness in practice

In our experiments on Cifar-10 the best performing architecture was a MLP Mixer with a single block of mixing. We measure the theoretical bounds with our framework at the start of training in Fig. 11, and at the end of training in Fig. 12.

We see that the last layer benefits from bounds that are quite tight (around 30% to 50%) whereas the tightness drops for the deeper layers in MLP blocks (less than 10%). The explanation is that the bound

$$\left\| \frac{\partial f_d(\theta_d, x)}{\partial \theta_d} \right\|_2 \leq K(f_d, \theta_d) X_{d-1}, \tag{28}$$

might be overly pessimistic, because of the $X_{d-1}$ term, especially since Cauchy-Schwartz inequality is a "best-case bound" (i.e assuming that all vector involved in the inner product are co-linear), and do not account for the possible orthogonality between $x_d$ and the cotangent vector. Nonetheless, with a ratio of 4% we see that the noise and the gradient norm are of similar amplitude as soon as

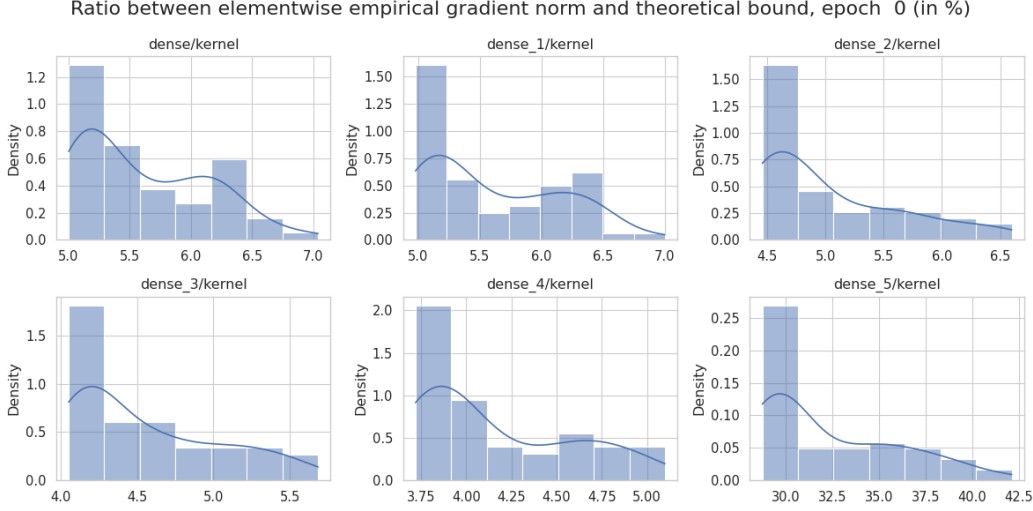

Figure 11: **MLP Mixer on Cifar-10, first epoch.**

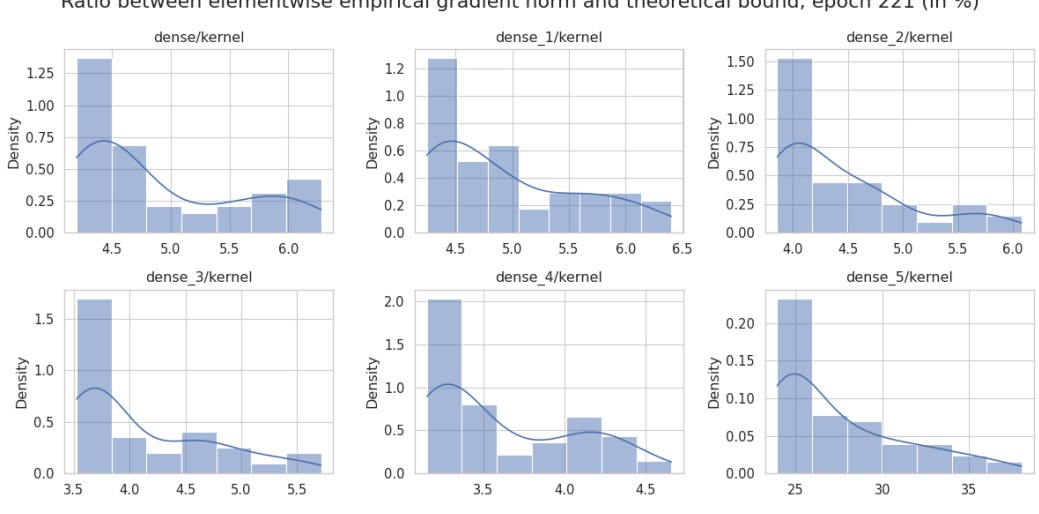

Figure 12: **MLP Mixer on Cifar-10, last epoch.**

| Dataset | Samples | Features | $\delta$ | Validation AUROC ↑ | | Wallclock Runtime (s) ↓ | |
|---|---|---|---|---|---|---|---|
| | | | | DP-SGD | Clipless DP-SGD | DP-SGD | Clipless DP-SGD |
| ALOI | 39,627 | 27 | $10^{-5}$ | **56.5** | 56.2 | 159.1 | **11.3** |
| campaign | 32,950 | 62 | $10^{-5}$ | **90.0** | 82.2 | 155.6 | **11.8** |
| celeba | 162,079 | 39 | $10^{-6}$ | **96.6** | 96.5 | 41.1 | **34.6** |
| census | 239,428 | 500 | $10^{-6}$ | **93.3** | 92.5 | 820.0 | **79.8** |
| donors | 495,460 | 10 | $10^{-6}$ | 100.0 | **100.0** | 257.9 | **90.2** |
| magic | 15,216 | 10 | $10^{-5}$ | **90.7** | 89.7 | 89.9 | **56.0** |
| shuttle | 39,277 | 9 | $10^{-5}$ | 98.3 | **99.4** | 11.1 | **6.8** |
| skin | 196,045 | 3 | $10^{-6}$ | **100.0** | 99.8 | 54.2 | **40.2** |
| yeast | 1,187 | 8 | $10^{-4}$ | 66.8 | **75.1** | 22.1 | **5.6** |

Table 7: Best validation AUROC values (in %) for models trained under $(\epsilon, \delta)$-DP privacy with $\epsilon = 1$, with DP-SGD and Clipless DP-SGD, on binary classification tasks of tabular data from Adbench datasets Han et al. (2022). We use a random 80/20% stratified split into train/val.

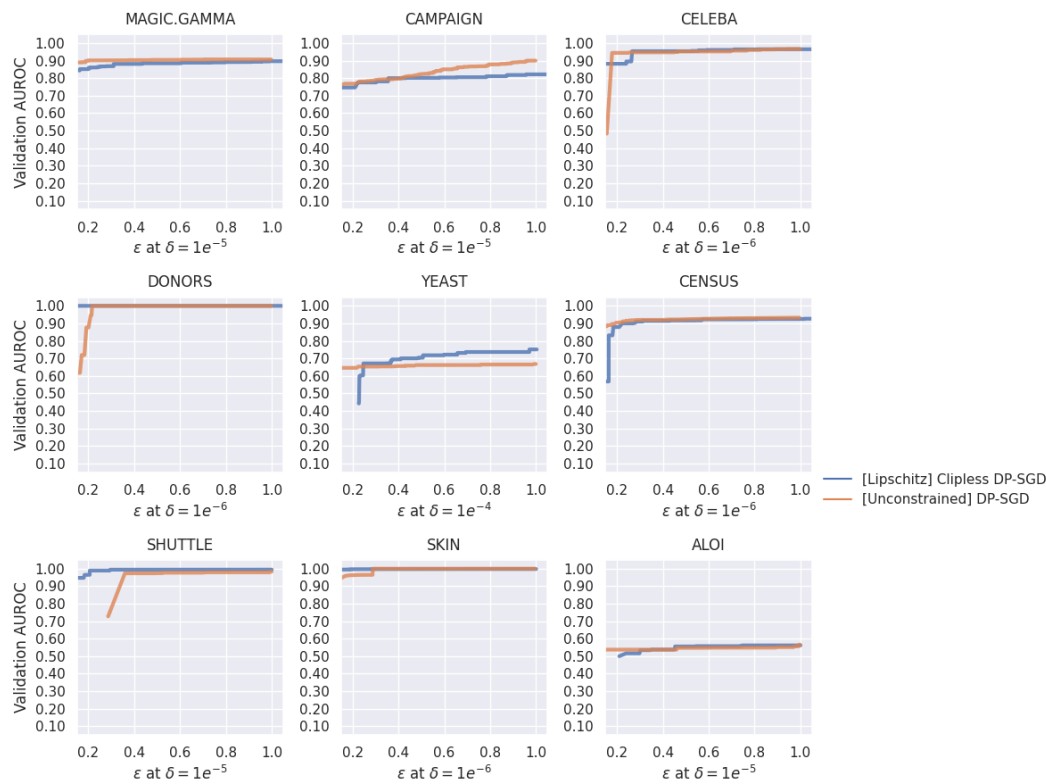

Figure 13: **Pareto front of AUROC values** on tabular data.

the batch size exceed $1/0.04 = 25$ examples, which is clearly our case with batch sizes that easily exceed $2,500$ on Cifar-10.

# D EXPERIMENTAL SETUP

## D.1 TABULAR DATA

Instead of monitoring the median result, we can also look for the best one in each sweep, along with the average runtime. The results are also report graphically in Fig. 13. The hyper-parameters are reported in Table 8.

| Hyper-Parameter | Minimum Value | Maximum Value |
|---|---|---|
| Input Clipping | None | |
| Batch Size | 512 | $10^4$ |
| Loss Gradient Clipping | automatic (90%-th percentile) | |
| $\tau$ (BCE) | $10^{-2}$ | 100 |
| Learning Rate | 0.00001 | 1.0 |

Table 8: **Hyper-parameters of Clipless DP-SGD for the Tabular experiment.**

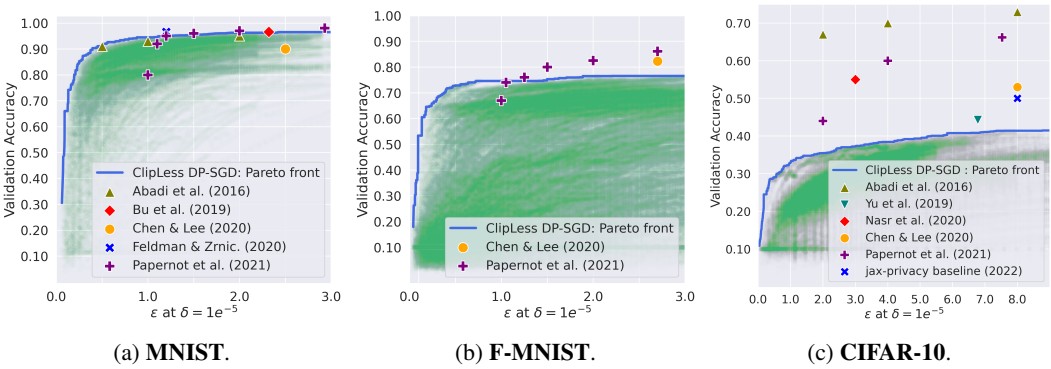

(a) **MNIST**.  (b) **F-MNIST**.  (c) **CIFAR-10**.

Figure 14: **Our framework paints a clearer picture of the privacy/utility trade-off.** We trained models in an "out of the box setting" (no pre-training, no data augmentation and no handcrafted features) on multiple tasks. Each green dot corresponds to an (**accuracy**, $\epsilon$) pair from an epoch of one of the runs, while the blue line is the Pareto front (convex hull of all the dots). While our results align with the baselines presented in other frameworks, we recognize the importance of domain-specific engineering. In this regard, we find the innovations introduced in Papernot et al. (2021); Tramer & Boneh (2021); De et al. (2022) and references therein highly relevant. These advancements demonstrate compatibility with our framework and hold potential for future integration.

## D.2 Pareto fronts

We rely on Bayesian optimization Snoek et al. (2012) with Hyper-band Li et al. (2017) heuristic for early stopping. The influence of some hyperparameters has to be highlighted to facilitate training with our framework, therefore we provide a table that provides insights into the effects of principal hyperparameters in Figure 15. Most hyper-parameters extend over different scales (such as the learning rate), so they are sampled according to log-uniform distribution, to ensure fair covering of the search space. Additionnaly, the importance of the softmax cross-entropy temperature $\tau$ has been demonstrated in previous work Béthune et al. (2022).

| Hyperparameter Tuning | Influence on utility | Influence on privacy leakage per step |
|---|---|---|
| Increasing Batch Size | Beneficial: decreases the sensitivity. | Detrimental: reduces the privacy amplification by subsampling. |
| Loss Gradient Clipping | Beneficial: tighter sensitivity bounds. Detrimental: biases the direction of the gradient. | No influence |
| Clipping Input Norms | Detrimental: destroy information, but may increases generalization | No influence |

Figure 15: **Hyperparameter table:** Here, we give insights on the influence of some hyper-parameters on utility and privacy.

### D.2.1 Hyperparameters configuration for MNIST

Experiments are run on NVIDIA GeForce RTX 3080 GPUs. The losses we optimize are either the Multiclass Hinge Kantorovich Rubinstein loss or the $\tau - \text{CCE}$.

| Hyper-Parameter | Minimum Value | Maximum Value |
|---|---|---|
| Input Clipping | $10^{-1}$ | 1 |
| Batch Size | 512 | $10^4$ |
| Loss Gradient Clipping | automatic (90%-th percentile) | |
| $\alpha$ (HKR) | $10^{-2}$ | $2,0 \times 10^3$ |
| $\tau$ (CCE) | $10^{-2}$ | $1,8 \times 10^1$ |

The sweeps were run with MLP or ConvNet architectures yielding the results presented in Figure 14a.

### D.2.2 Hyperparameters configuration for FASHION-MNIST

Experiments are run on NVIDIA GeForce RTX 3080 GPUs. The losses we optimize are the Multiclass Hinge Kantorovich Rubinstein loss, the $\tau - \text{CCE}$ or the K-CosineSimilarity custom loss function.

| Hyper-Parameter | Minimum Value | Maximum Value |
|---|---|---|
| Input Clipping | $10^{-1}$ | 1 |
| Batch Size | $5.0 \times 10^3$ | $10^4$ |
| Loss Gradient Clipping | automatic (90%-th percentile) | |
| $\alpha$ (HKR) | $10^{-2}$ | $2.0 \times 10^3$ |
| $\tau$ (CCE) | $10^{-2}$ | $4.0 \times 10^1$ |
| $K$ (K-CS) | $10^{-2}$ | $1.0$ |

A simple ConvNet architecture, in the spirit of VGG (but with Lipschitz constraints), was chosen to run all sweeps yielding the results we present in Figure 14b.

### D.2.3 Hyperparameters configuration for CIFAR-10

Experiments are run on NVIDIA GeForce RTX 3080 or 3090 GPUs. The losses we optimize are the Multiclass Hinge Kantorovich Rubinstein loss, the $\tau - \text{CCE}$ or the K-CosineSimilarity custom loss function. They yield the results of Figure 14c.

| Hyper-Parameter | Minimum Value | Maximum Value |
|---|---|---|
| Input Clipping | $10^{-2}$ | 1 |
| Batch Size | 512 | $10^4$ |
| Loss Gradient Clipping | automatic (90%-th percentile) | |
| $\alpha$ (HKR) | $10^{-2}$ | $2.0 \times 10^3$ |
| $\tau$ (CCE) | $10^{-3}$ | $3.2 \times 10^1$ |
| $K$ (K-CS) | $10^{-2}$ | $1.0$ |

The sweeps have been done on various architectures such as Lipschitz VGGs, Lipschitz ResNets and Lipschitz MLP_Mixer. We can also break down the results per architecture, in figure 16. The MLP_Mixer architecture seems to yield the best results. This architecture is exactly GNP since the orthogonal linear transformations are applied on disjoint patches. To the contrary, VGG and Resnets are based on RKO convolutions which are not exactly GNP. Hence those preliminary results are compatible with our hypothesis that GNP layers should improve performance. Note that these results are expected to change as the architectures are further improved. It is also dependant of the range chosen for hyper-parameters. We do not advocate for the use of an architecture over another, and we believe many other innovations found in literature should be included before settling the question definitively.

For the vanilla implementation of DP-SGD we rely on Opacus library. We use the default configuration from the official tutorial on Cifar-10, and we define the hyper-parameters for the grid search below:

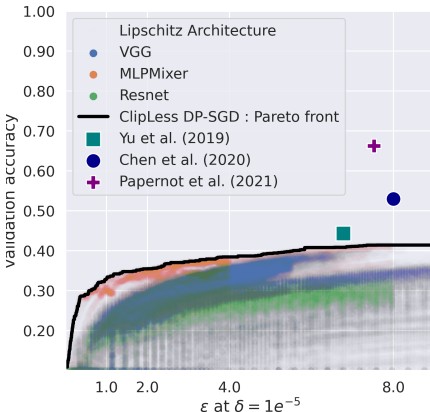

Figure 16: Accuracy/Privacy tradeoff on Cifar-10, split down per architecture used. While some architectures seems to perform better than others, we don't advocate for the use of one over another. The results may not translate to all datasets, and may be highly dependant on the range chosen for hyper-parameters. While this figure provides valuable insights, identifying the best architecture is left for future works.

| Hyper-Parameter | Minimum Value | Maximum Value |
|:---:|:---:|:---:|
| Input | RGB standardized | |
| Batch Size | 500 | 5000 |
| Noise Multiplier | Automatic | |
| Epochs | 1 | 400 |
| Maximum Gradient norm | 0.01 | 100 |
| Learning rate | 0.00001 | 1 |

### D.3 CONFIGURATION OF THE "SPEED" EXPERIMENT

We detail below the environment version of each experiment, together with Cuda and Cudnn versions. We rely on a machine with 32GB RAM and a NVIDIA Quadro GTX 8000 graphic card with 48GB memory. The GPU uses driver version 495.29.05, cuda 11.5 (October 2021) and cudnn 8.2 (June 7, 2021). We use Python 3.8 environment.

- For Jax, we used jax 0.3.17 (Aug 31, 2022) with jaxlib 0.3.15 (July 23, 2022), flax 0.6.0 (Aug 17, 2022) and optax 1.4.0 (Nov 21, 2022).
- For Tensorflow, we used tensorflow 2.12 (March 22, 2023) with tensorflow_privacy 0.7.3 (September 1, 2021).
- For Pytorch, we used Opacus 1.4.0 (March 24, 2023) with Pytorch (March 15, 2023).
- For lip-dp we used deel-lip 1.4.0 (January 10, 2023) on Tensorflow 2.8 (May 23, 2022).

For this benchmark, we used among the most recent packages on pypi. However the latest version of tensorflow privacy could not be forced with pip due to broken dependencies. This issue arise in clean environments such as the one available in google colaboratory.

### D.4 COMPATIBILITY WITH LITERATURE IMPROVEMENTS

Many method from the state of the art are based on improvement over the DP-SGD baseline. In this section we will review these improvements and check the compatibility with our approach.

**Adaptive clipping.** As discussed in 3.1, or approach can benefit from adaptive clipping. Similarly this can be done in a more efficient way.

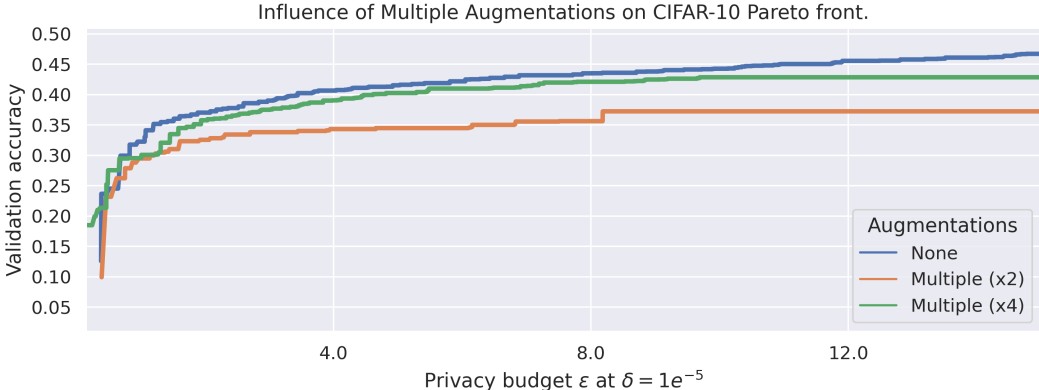

Figure 17: **Pareto front on Cifar-10 with the multiple augmentations trick of De et al. (2022)**.

**Multiple augmentations**    We tested adding "multiple augmentation" introduced by De et al. (2022) on our approach but it did not yield improvements (see Appendix 17). This is probably due to the fact that we train our architecture from scratch: doing so require an architecture that requires a minimum amount of steps to converge. Such architecture are too small and not able to learn an augmented dataset as those are more prone to under-fitting. Another reason is that in vanilla DP-SGD the elementwise gradients of each augmentation can be averaged *before* the clipping operation, which has no consequences on the overall sensitivity of the gradient step. Whereas in our case, the average is computed *after* the loss gradient clipping.

**Fine tuning a pretrained backbone:**    Clipless DP-SGD can work with a pre-trained backbone, however, the fine-tuned layers must be 1-lipschitz. This can yield 2 approaches:

- **Using a 1-lipschitz backbone:** We can then fine tune the whole network and benefits from pre-training. Unfortunately there is no 1-lipschitz backbone available for the moment.
- **Using an unconstrained backbone:** Our approach can work with an unconstrained feature extractor using the following protocol: 1. the $n$ last layers are dropped and replaced with a 1-Lipschitz classification network. 2. An input clipping layer is added at the beginning of the classification head to ensure that the inputs are bounded. This approach was tested using a MobilenetV2 backbone. Results are reported in Figure 18.

### D.5    DROP-IN REPLACEMENT WITH LIPSCHITZ NETWORKS IN VANILLA DPSGD

To highlight the importance of the tight sensitivity bounds $\Delta_d$ obtained by our framework, we perform an ablation study by optimizing GNP networks using "vanilla" DP-SGD (with clipping), in Figure 19a and 19b.

Thanks to the gradient clipping of DP-SGD (see Algorithm 4), Lipschitz networks can be readily integrated in traditional DP-SGD algorithm with gradient clipping. The PGD algorithm is not mandatory: the back-propagation can be performed within the computation graph through iterations of Björck algorithm (used in RKO convolutions). This does not benefit from any particular speed-up over conventional networks - quite to the contrary there is an additional cost incurred by enforcing Lipschitz constraints in the graph. Some layers of deel-lip library have been recoded in Jax/Flax, and the experiment was run in Jax, since Tensorflow was too slow.

We use use the Total Amount of Noise (TAN) heuristic introduced in Sander et al. (2022) to heuristically tune hyper-parameters jointly. This ensures fair covering of the Pareto front.

### D.6    EXTENDED LIMITATIONS

The main weakness of our approach is that it crucially rely on accurate computation of the sensitivity $\Delta$. This task faces many challenges in the context of differential privacy: floating point

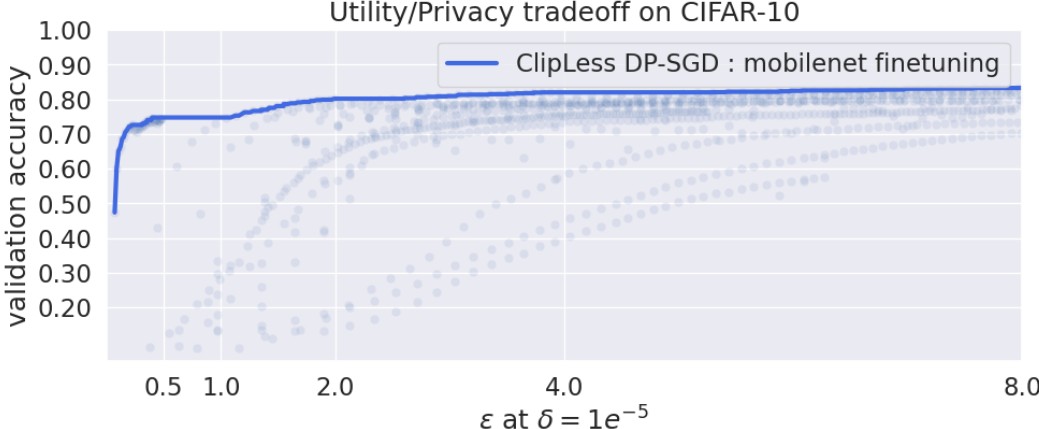

Figure 18: **Finetuning of a MobilenetV2 with Clipless DP-SGD.** A Lipschitz MLP (1 or 2 layers, and GroupSort activation) is trained using the backbone as a feature extractor. The backbone is not fine-tuned as it is not Lipschitz. Therefore a plateau exists, depending on the quality of the feature extractor.

---

**Algorithm 4** Differentially Private Stochastic Gradient Descent : **DP-SGD**

---

**Input**: Neural network architecture $f(\cdot, \cdot)$
**Input**: Initial weights $\theta_0$, learning rate scheduling $\eta_t$, number of steps $N$, noise multiplier $\sigma$, L2 clipping value $C$ .

1: **repeat**
2:     **for all** $1 \leq t \leq N - 1$ **do**
3:         Sample a batch
$$\mathcal{B}_t = (x_1, y_1), (x_2, y_2), \ldots, (x_b, y_b).$$
4:         Create microbatches, compute and clip the per-sample gradient of cost function:
$$\tilde{g}_{t,i} := \min(C, \|\nabla_{\theta_t}\mathcal{L}(\hat{y}_i, y_i)\|) \frac{\nabla_{\theta_t}\mathcal{L}(\hat{y}_i, y_i)}{\|\nabla_{\theta_t}\mathcal{L}(\hat{y}_i, y_i)\|}.$$
5:         Perturb each microbatch with carefully chosen noise distribution $b \sim \mathcal{N}(0, \sigma C)$ :
$$\hat{g}_{t,i} \leftarrow \tilde{g}_{t,i} + b_i.$$
6:         Perform projected gradient step:
$$\theta_{t+1} \leftarrow \Pi(\theta_t - \eta_t\hat{g}_{t,i}).$$
7:     **end for**
8: **until** privacy budget $(\epsilon, \delta)$ has been reached.

---

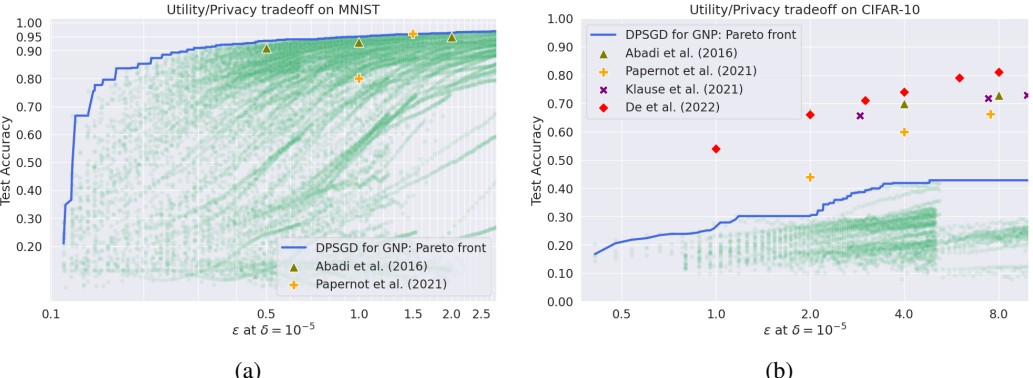

(a)                                                                 (b)

Figure 19: **Privacy/utility trade-off for Gradient Norm Preserving networks** trained under "vanilla" DP-SGD (**with** gradient clipping). Each green dot corresponds to a single epoch of one of the runs. Trajectories that end abruptly are due to the automatic early stopping of unpromising runs. Note that clipping + orthogonalization have a high runtime cost, which severely limits the number of epochs reported.

arithmetic is not associative, and summation order can a have dramatic consequences regarding numerical stability Goldberg (1991). This is further amplified on the GPUs, where some operations are intrinsically non deterministic Jooybar et al. (2013). This well known issue is already present in vanilla DP-SGD algorithm. Our framework adds an additional point of failure: the upper bound of spectral Jacobian must be computed accurately. Hence Power Iteration must be run with sufficiently high number of iterations to ensure that the projection operator $\Pi$ works properly. The $(\epsilon, \delta)$-DP certificates only hold under the hypothesis that all computations are correct, as numerical errors can induce privacy leakages. Hence we check empirically the effective norm of the gradient in the training loop at the end of each epoch. No certificate violations were reported during ours experiments, which suggests that the numerical errors can be kept under control.

## E  PROOFS OF GENERAL RESULTS

This section contains the proofs of results that are either informally presented in the main paper, either formally defined in section A.5.

### E.1  VARIANCE OF THE GRADIENT

The result mentionned in section 3.1 is based on the following result, directly adapted from a classical result on concentration inequalities.

**Corollary 2.  Concentration of stochastic gradient around its mean.** *Assume the samples $(x, y)$ are i.i.d and sampled from an arbitrary distribution $\mathcal{D}$. We introduce the R.V $g = \nabla_\theta \mathcal{L}(x, y)$ which is a function of the sample $(x, y)$, and its expectation $\bar{g} = \mathbb{E}_{(x,y)\sim\mathcal{D}}[\nabla_\theta \mathcal{L}(x, y)]$. Then for all $u \geq \frac{2}{\sqrt{b}}$ the following inequality hold:*

$$\mathbb{P}(\|\frac{1}{b}\sum_{i=1}^{b} g_i - \bar{g}\| > uK) \leq \exp\left(-\frac{\sqrt{b}}{8}(u - \frac{2}{\sqrt{b}})^2\right). \tag{29}$$

*Proof.* The result is an immediate consequence of Example 6.3 p167 in Boucheron et al. (2013). We apply the theorem with the centered variable $X_i = \frac{1}{b}(g_i - \bar{g})$ that fulfills condition $\|X_i\| \leq \frac{c_i}{2}$ with $c_i = \frac{4K}{b}$ since $\|g_i\| \leq K$. Then for every $t \geq \frac{2K}{\sqrt{b}}$ we have:

$$\mathbb{P}(\|\frac{1}{b}\sum_{i=1}^{b} g_i - \bar{g}\| > t) \leq \exp\left(-\frac{\sqrt{b}}{8K^2}(t - \frac{2K}{\sqrt{b}})^2\right). \tag{30}$$

We conclude with the change of variables $u = \frac{t}{K}$.                                                       □

The informal Theorem 1 requires some tools that we introduce below.

**Additionnal hypothesis for GNP networks.**   We introduce convenient assumptions for the purpose of obtaining tight bounds in Algorithm 2.

**Assumption 1** (Bounded biases). *We assume there exists $B > 0$ such that that for all biases $b_d$ we have $\|b_d\| \le B$. Observe that the ball $\{\|b\|_2 \le B\}$ of radius $B$ is a **convex** set.*

**Assumption 2** (Zero preserving activation). *We assume that the activation fulfills $\sigma(\mathbf{0}) = \mathbf{0}$. When $\sigma$ is S-Lipschitz this implies $\|\sigma(x)\| \le S\|x\|$ for all $x$. Examples of activations fulfilling this constraints are ReLU, Groupsort, GeLU, ELU, tanh. However it does not work with sigmoid or softplus.*

We also propose the assumption 3 for convenience and exhaustivity.

**Assumption 3** (Bounded activation). *We assume it exists $G > 0$ such that for every $x \in \mathcal{X}$ and every $1 \le d \le D + 1$ we have:*

$$\|h_d\| \le G \quad and \quad \|z_d\| \le G. \tag{31}$$

*Note that this assumption is implied by requirement 2, assumption 1-2, as illustrated in proposition 3.*

In practice assumption 3 can be fulfilled with the use of input clipping and bias clipping, bounded activation functions, or layer normalization. This assumption can be used as a "shortcut" in the proof of the main theorem, to avoid the "propagation of input bounds" step.

### E.3   MAIN RESULT

We rephrase in a rigorous manner the informal theorem of section 3.1. The proofs are given in section B. In order to simplify the notations, we use $X := X_0$ in the following.

**Proposition 3.   Norm of intermediate activations.** *Under requirement 2, assumptions 1-2 we have:*

$$\|h_t\| \le S\|z_t\| \le \begin{cases} (US)^t \left(X - \frac{SB}{1-SU}\right) + \frac{SB}{1-SU} & if\, US \ne 1, \\ SX + tSB & otherwise. \end{cases} \tag{32}$$

*In particular if there are no biases, i.e if $B = 0$, then $\|h_t\| \le S\|z_t\| \le SX$ .*

Proposition 3 can be used to replace assumption 3.

**Proposition 4.   Lipschitz constant of dense Lipschitz networks with respect to parameters.** *Let $f(\cdot, \cdot)$ be a Lipschitz neural network. Under requirement 2, assumptions 1-2 we have for every $1 \le t \le T + 1$:*

$$\|\frac{\partial f(\theta, x)}{\partial b_t}\|_2 \le (SU)^{T+1-t}, \tag{33}$$

$$\|\frac{\partial f(\theta, x)}{\partial W_t}\|_2 \le (SU)^{T+1-t}\|h_{t-1}\|. \tag{34}$$

*In particular, for every $x \in \mathcal{X}$, the function $\theta \mapsto f(\theta, x)$ is Lipschitz bounded.*

Proposition 4 suggests that the scale of the activation $\|h_t\|$ must be kept under control for the gradient scales to distribute evenly along the computation graph. It can be easily extended to a general result on the *per sample* gradient of the loss, in theorem 1.

**Theorem 1.   Bounded loss gradient for dense Lipschitz networks.** *Assume the predictions are given by a Lipschitz neural network $f$:*

$$\hat{y} := f(\theta, x). \tag{35}$$

*Under requirements 1-2, assumptions 1-2, there exists a $K > 0$ for all $(x, y, \theta) \in \mathcal{X} \times \mathcal{Y} \times \Theta$ the loss gradient is bounded:*

$$\|\nabla_\theta \mathcal{L}(\hat{y}, y)\|_2 \le K. \tag{36}$$

*Let $\alpha = SU$ be the maximum spectral norm of the Jacobian between two consecutive layers.*

**If $\alpha = 1$ then we have:**

$$K = \mathcal{O}\left(LX + L\sqrt{T} + LSX\sqrt{T} + L\sqrt{BX}ST + LBST^{3/2}\right). \tag{37}$$

*The case $S = 1$ is of particular interest since it covers most activation function (i.e ReLU, GroupSort):*

$$K = \mathcal{O}\left(L\sqrt{T} + LX\sqrt{T} + L\sqrt{BX}T + LBT^{3/2}\right). \tag{38}$$

*Further simplification is possible if we assume $B = 0$, i.e a network without biases:*

$$K = \mathcal{O}\left(L\sqrt{T}(1 + X)\right). \tag{39}$$

**If $\alpha > 1$ then we have:**

$$\|\nabla_\theta \mathcal{L}(\hat{y}, y)\|_2 = \mathcal{O}\left(L\frac{\alpha^T}{\alpha - 1}\left(\sqrt{T}(\alpha X + SB) + \frac{\alpha(SB + \alpha)}{\sqrt{\alpha^2 - 1}}\right)\right). \tag{40}$$

*Once again $B = 0$ (network with no bias) leads to useful simplifications:*

$$\|\nabla_\theta \mathcal{L}(\hat{y}, y)\|_2 = \mathcal{O}\left(L\frac{\alpha^{T+1}}{\alpha - 1}\left(\sqrt{T}X + \frac{\alpha}{\sqrt{\alpha^2 - 1}}\right)\right). \tag{41}$$

*We notice that when $\alpha \gg 1$ there is an **exploding gradient** phenomenon where the upper bound become vacuous.*

**If $\alpha < 1$ then we have:**

$$\|\nabla_\theta \mathcal{L}(\hat{y}, y)\|_2 = \mathcal{O}\left(L\alpha^T\left(X\sqrt{T} + \frac{1}{(1 - \alpha^2)}\left(\sqrt{\frac{XSB}{\alpha^T}} + \frac{SB}{\sqrt{(1 - \alpha)}}\right)\right) + \frac{L}{(1 - \alpha)\sqrt{1 - \alpha}}\right). \tag{42}$$

*For network without biases we get:*

$$\|\nabla_\theta \mathcal{L}(\hat{y}, y)\|_2 = \mathcal{O}\left(L\alpha^T X\sqrt{T} + \frac{L}{\sqrt{(1 - \alpha)^3}}\right). \tag{43}$$

*The case $\alpha \ll 1$ is a **vanishing gradient** phenomenon where $\|\nabla_\theta \mathcal{L}(\hat{y}, y)\|_2$ is now independent of the depth $T$ and of the input scale $X$.*

*Proof.* The control of gradient implicitly depend on the scale of the output of the network at every layer, hence it is crucial to control the norm of each activation.

**Lemma 1** (Bounded activations). *If $US \neq 1$ for every $1 \leq t \leq T + 1$ we have:*

$$\|z_t\| \leq U^t S^{t-1}\left(X - \frac{SB}{1 - SU}\right) + \frac{B}{1 - SU}. \tag{44}$$

*If $US = 1$ we have:*

$$\|z_t\| \leq X + tB. \tag{45}$$

*In every case we have $\|h_t\| \leq S\|z_t\|$.*

*Lemma proof.* From assumption 2, if we assume that $\sigma$ is $S$-Lipschitz, we have:

$$\|h_t\| = \|\sigma(z_t)\| = \|\sigma(z_t) - \sigma(\mathbf{0})\| \leq \mathbf{S}\|\mathbf{z_t}\|. \tag{46}$$

Now, observe that:

$$\|z_{t+1}\| = \|W_{t+1}h_t + b_{t+1}\| \leq \|W_{t+1}\|\|h_t\| + \|b_{t+1}\| \leq US\|z_t\| + B. \tag{47}$$

Let $u_1 = UX + B$ and $u_{t+1} = SUu_t + B$ be a linear recurrence relation. The translated sequence $u_t - \frac{B}{1-SU}$ is a geometric progression of ratio $SU$, hence $u_t = (SU)^{t-1}(UX + B - \frac{B}{1-SU}) + \frac{B}{1-SU}$. Finally we conclude that by construction $\|z_t\| \leq u_t$. $\blacksquare$

The activation jacobians can be bounded by applying the chainrule. The recurrence relation obtained is the one automatically computed with back-propagation.

**Lemma 2** (Bounded activation derivatives). *For every $T + 1 \geq s \geq t \geq 1$ we have:*

$$\|\frac{\partial z_s}{\partial z_t}\| \leq (SU)^{s-t}. \tag{48}$$

*Lemma proof.* The chain rule expands as:

$$\frac{\partial z_s}{\partial z_t} = \frac{\partial z_s}{\partial h_{s-1}} \frac{\partial h_{s-1}}{\partial z_{s-1}} \frac{\partial z_{s-1}}{\partial z_t}. \tag{49}$$

From Cauchy-Schwartz inequality we get:

$$\|\frac{\partial z_s}{\partial z_t}\| \leq \|\frac{\partial z_s}{\partial h_{s-1}}\| \cdot \|\frac{\partial h_{s-1}}{\partial z_{s-1}}\| \cdot \|\frac{\partial z_{s-1}}{\partial z_t}\|. \tag{50}$$

Since $\sigma$ is $S$-Lipschitz, and $\|W_s\| \leq U$, and by observing that $\|\frac{\partial z_t}{\partial z_t}\| = 1$ we obtain by induction that:

$$\|\frac{\partial h_s}{\partial h_t}\| \leq (SU)^{s-t}. \tag{51}$$

∎

The derivatives of the biases are a textbook application of the chainrule.

**Lemma 3** (Bounded bias derivatives). *For every $t$ we have:*

$$\|\nabla_{b_t} \mathcal{L}(\hat{y}, y)\| \leq L(SU)^{T+1-t}. \tag{52}$$

*Lemma proof.* The chain rule yields:

$$\nabla_{b_t} \mathcal{L}(\hat{y}, y) = (\nabla_{\hat{y}} \mathcal{L}(\hat{y}, y)) \frac{\partial z_{T+1}}{\partial z_t} \frac{\partial z_t}{\partial b_t}. \tag{53}$$

Hence we have:

$$\|\nabla_{b_t} \mathcal{L}(\hat{y}, y)\| = \|\nabla_{\hat{y}} \mathcal{L}(\hat{y}, y)\| \cdot \|\frac{\partial z_{T+1}}{\partial z_t}\| \cdot \|\frac{\partial z_t}{\partial b_t}\|. \tag{54}$$

We conclude with Lemma 2 that states $\|\frac{\partial z_{T+1}}{\partial z_t}\| \leq (US)^{T+1-t}$, with requirement 1 that states $\|\nabla_{\hat{y}} \mathcal{L}(\hat{y}, y)\| \leq L$ and by observaing that $\|\frac{\partial z_t}{\partial b_t}\| = 1$. ∎

We can now bound the derivative of the affine weights:

**Lemma 4** (Bounded weight derivatives). *For every $T + 1 \geq t \geq 2$ we have:*

$$\|\nabla_{W_t} \mathcal{L}(\hat{y}, y)\| \leq L(SU)^T \left( X - \frac{SB}{1 - SU} \right) + L(SU)^{T+1-t} \frac{SB}{1 - SU} \text{ when } SU \neq 1, \tag{55}$$

$$\|\nabla_{W_t} \mathcal{L}(\hat{y}, y)\| \leq LS \left( X + (t-1)B \right) \text{ when } SU = 1. \tag{56}$$

$$\tag{57}$$

*In every case:*

$$\|\nabla_{W_1} \mathcal{L}(\hat{y}, y)\| \leq L(SU)^T X. \tag{58}$$

*Lemma proof.* We proceed like in the proof of Lemma 3 and we get:

$$\|\nabla_{W_t} \mathcal{L}(\hat{y}, y)\| \leq \|\nabla_{\hat{y}} \mathcal{L}(\hat{y}, y)\| \cdot \|\frac{\partial z_{T+1}}{\partial z_t}\| \cdot \|\frac{\partial z_t}{\partial W_t}\|. \tag{59}$$

Which then yields:

$$\|\nabla_{W_t} \mathcal{L}(\hat{y}, y)\| \leq L(SU)^{T+1-t} \cdot \|\frac{\partial z_t}{\partial W_t}\|. \tag{60}$$

Now, for $T + 1 \geq t \geq 1$, according to Lemma 1 we either have:

$$\|\frac{\partial z_t}{\partial W_t}\| \leq \|h_{t-1}\| \leq S\|z_{t-1}\| = (SU)^{t-1}\left(X - \frac{SB}{1 - SU}\right) + \frac{SB}{1 - SU}, \tag{61}$$

or, when $US = 1$:

$$\|\frac{\partial z_t}{\partial W_t}\| \leq \|h_{t-1}\| = S\|z_{t-1}\| = SX + (t-1)SB \text{ if } t \geq 2, \tag{62}$$

$$\|\frac{\partial z_t}{\partial W_t}\| \leq X \text{ otherwise.} \tag{63}$$

∎

Now, the derivatives of the loss with respect to each type of parameter (i.e $W_t$ or $b_t$) are know, and they can be combined to retrieve the overall gradient vector.

$$\theta = \{(W_1, b_1), (W_2, b_2), \ldots (W_{T+1}, b_{T+1})\}. \tag{64}$$

We introduce $\alpha = SU$.

**Case $\alpha = 1$.** The resulting norm is given by the series:

$$\|\nabla_\theta \mathcal{L}(\hat{y}, y)\|_2^2 = \sum_{t=1}^{T+1} \|\nabla_{b_t}\mathcal{L}(\hat{y}, y)\|_2^2 + \|\nabla_{W_t}\mathcal{L}(\hat{y}, y)\|_2^2 \tag{65}$$

$$\leq L^2\left((1 + X^2) + \sum_{t=2}^{T+1}(1 + (SX + (t-1)SB)^2)\right) \tag{66}$$

$$\leq L^2\left(1 + X^2 + \sum_{u=1}^{T}(1 + (SX + uSB)^2)\right) \tag{67}$$

$$\leq L^2\left(1 + X^2 + \sum_{u=1}^{T}(1 + S^2(X^2 + +2uBX + u^2B^2))\right) \tag{68}$$

$$\leq L^2\left(1 + X^2 + T(1 + S^2X^2) + S^2BXT(T + 1) + S^2B^2\frac{T(T + 1)(2T + 1)}{6}\right). \tag{69}$$

Finally:

$$\|\nabla_\theta \mathcal{L}(\hat{y}, y)\|_2 = \mathcal{O}(L\sqrt{X^2 + T + TS^2X^2 + BS^2XT^2 + B^2S^2T^3}) \tag{70}$$

$$= \mathcal{O}\left(LX + L\sqrt{T} + LSX\sqrt{T} + L\sqrt{BX}ST + LBST^{3/2}\right). \tag{71}$$

This upper bound depends (asymptotically) linearly of $L, X, S, B, T^{3/2}$, when other factors are kept fixed to non zero value.

**Case** $\alpha \neq 1$. We introduce $\beta = \frac{SB}{1-\alpha}$.

$$\|\nabla_\theta \mathcal{L}(\hat{y}, y)\|_2^2 = \sum_{t=1}^{T+1} \|\nabla_{b_t} \mathcal{L}(\hat{y}, y)\|_2^2 + \|\nabla_{W_t} \mathcal{L}(\hat{y}, y)\|_2^2 \tag{72}$$

$$\leq L^2 \left( \alpha^{2T} \sum_{t=1}^{T+1} (((X - \beta) + \alpha^{1-t}\beta)^2 + \alpha^{2-2t}) \right) \tag{73}$$

$$\leq L^2 \alpha^{2T} \left( \sum_{u=0}^{T} (((X - \beta)^2 + 2(X - \beta)\alpha^{-u}\beta + \alpha^{-2u}\beta^2) + \alpha^{-2u}) \right) \tag{74}$$

$$\leq L^2 \alpha^{2T} \left( (T+1)(X - \beta)^2 + 2(X - \beta)\beta \sum_{u=0}^{T} \alpha^{-u} + (\beta^2 + 1) \sum_{u=0}^{T} \alpha^{-2u} \right) \tag{75}$$

$$\leq L^2 \alpha^{2T} \left( (T+1)(X - \beta)^2 + 2(X - \beta)\beta \frac{\alpha - (\frac{1}{\alpha})^T}{\alpha - 1} + (\beta^2 + 1)\frac{\alpha^2 - (\frac{1}{\alpha^2})^T}{\alpha^2 - 1} \right). \tag{76}$$

Finally:

$$\|\nabla_\theta \mathcal{L}(\hat{y}, y)\|_2 \leq L\alpha^T \sqrt{(T+1)(X - \beta)^2 + 2(X - \beta)\beta \frac{\alpha - (\frac{1}{\alpha})^T}{\alpha - 1} + (\beta^2 + 1)\frac{\alpha^2 - (\frac{1}{\alpha^2})^T}{\alpha^2 - 1}}. \tag{77}$$

Now, the situation is a bit different for $\alpha < 1$ and $\alpha > 1$. One case corresponds to exploding gradient, and the other to vanishing gradient.

When $\alpha < 1$ we necessarily have $\beta > 0$, hence we obtain a crude upper-bound:

$$\|\nabla_\theta \mathcal{L}(\hat{y}, y)\|_2 = \mathcal{O}\left( L\alpha^T \left( X\sqrt{T} + \frac{1}{(1 - \alpha^2)} \left( \sqrt{\frac{XSB}{\alpha^T}} + \frac{SB}{\sqrt{(1 - \alpha)}} \right) \right) + \frac{L}{(1 - \alpha)\sqrt{1 - \alpha}} \right). \tag{78}$$

Once again $B = 0$ (network with no bias) leads to useful simplifications:

$$\|\nabla_\theta \mathcal{L}(\hat{y}, y)\|_2 = \mathcal{O}\left( L\alpha^T X\sqrt{T} + \frac{L}{\sqrt{(1 - \alpha)^3}} \right). \tag{79}$$

This is a typical case of vanishing gradient since when $T \gg 1$ the upper bound does not depend on the input scale $X$ anymore.

Similarly, we can perform the analysis for $\alpha > 1$, which implies $\beta < 0$, yielding another bound:

$$\|\nabla_\theta \mathcal{L}(\hat{y}, y)\|_2 = \mathcal{O}\left( L\frac{\alpha^T}{\alpha - 1} \left( \sqrt{T}(\alpha X + SB) + \frac{\alpha(SB + \alpha)}{\sqrt{\alpha^2 - 1}} \right) \right). \tag{80}$$

Without biases we get:

$$\|\nabla_\theta \mathcal{L}(\hat{y}, y)\|_2 = \mathcal{O}\left( L\frac{\alpha^{T+1}}{\alpha - 1} \left( \sqrt{T}X + \frac{\alpha}{\sqrt{\alpha^2 - 1}} \right) \right). \tag{81}$$

We recognize an exploding gradient phenomenon due to the $\alpha^T$ term. $\qquad\square$

Propositions 3 and 4 were introduced for clarity. They are a simple consequence of the Lemmas 1-3-4 used in the proof of Theorem 1.

The informal theorem of section 3.1 is based on the bounds of theorem 1, that have been simplified. Note that the definition of network differs slightly: in definition 3 the activations and the affines layers are considered independent and indexed differently, while the theoretical framework merge them into $z_t$ and $h_t$ respectively, sharing the same index $t$. This is without consequences once we realize that if $K = U = S$ and $2T = D$ then $(US)^2 = \alpha^2 = K^2$ leads to $\alpha^{2T} = K^D$. The leading constant factors based on $\alpha$ value have been replaced by 1 since they do not affect the asymptotic behavior.

