# OpenReview forum: "DP-SGD Without Clipping: The Lipschitz Neural Network Way"
_ICLR.cc/2024/Conference — ICLR 2024 poster_

### Official Review · Reviewer_GMKM · 2023-11-01

**Soundness:** 3 good
**Presentation:** 2 fair
**Contribution:** 2 fair
**Rating:** 5
**Confidence:** 4

**Summary:**

This paper studies to train Lipschitz neural networks with DP, so as to avoid using per-sample gradient clipping. It gives theorems of the tradeoff between privacy and utility, as well as a Python package that efficiently implement the proposed algorithms. Empirical results on toy datasets show some promise of this new method.

**Strengths:**

Overall, the paper is well-polished and rigorous. I appreciate the careful layout of the requirements and proper introduction of Lipschitz literatures.

**Weaknesses:**

I have some concerns about the claims made in this work, in addition to some minor issues.

1. The significance of this work is based on the hypothesis that per-sample gradient clipping is inefficient. This has been emphasized in the abstract and in the paragraph below Definition 2, in which the authors wrote "Hyper-parameter search on the broad-range clipping value C is required...The computation of per-sample gradients is expensive". However, this is mistaken. DP-SGD does not need to tune the clipping value (see "Automatic clipping: Differentially private deep learning made easier and stronger" and "Normalized/Clipped SGD with Perturbation for Differentially Private Non-Convex Optimization"). DP-SGD/AdamW can be as fast and memory efficient as standard SGD, while maintaining the same DP accuracy. Many papers that use ghost clipping can achieve this (see "Differentially Private Optimization on Large Model at Small Cost", "Exploring the Limits of Differentially Private Deep Learning with Group-wise Clipping", "LARGE LANGUAGE MODELS CAN BE STRONG DIFFERENTIALLY PRIVATE LEARNERS", "Scalable and Efficient Training of Large Convolutional Neural Networks with Differential Privacy"), improving the speed from 1.7X slower than non-DP to 1.1X. It would be inappropriate if the authors only refer to the older works like Kifer's and Opacus.

2. Another issue is to claim per-sample clipping introduces bias and not to fully discuss the bias of Clipless DP-SGD. The gradient bias does exist, but may not be the reason of slow convergence. Furthermore, I believe Clipless DP-SGD also introduces some bias, in the form of network architecture by moving from regular ResNet to GNP ResNet. It would not be fair to discard the clipping by the current discussion in Theorem 1.

3. The empirical results in this work are not convincing, focusing only on toy datasets, restricted layer types (not including embedding layer that is vital to language model and vision transformer), and toy models (e.g. Figure 5, model size 2M). I wonder whether the new method can benefit from large scale pretraining?

**Questions:**

See Weaknesses.

---

> ### Author Response · Authors · 2023-11-16
> **Thank you for your careful reading**
>
> We appreciate your careful reading, and we agree that two claims in the paper can be better discussed and nuanced. We added an extended discussion in the appendix to which we refer from the introduction.
> Your third remark (on large-scale experiments) is addressed in the common response.
>
> ### Efficiency of gradient clipping
>
> > based on the hypothesis that per-sample gradient clipping is inefficient
>
> Thank you for the additional references. We agree that some recent work on gradient clipping proposes more efficient strategies than the ones implemented in tf-privacy, Opacus, and jax-privacy packages. To the best of our knowledge, these new techniques are not (yet) included in these libraries. In contrast, we propose a package ready to be used where the computational overhead is independent of the batch size which, to our knowledge, is a first for any DP training library.
>
> We added a comparative table between the different strategies of the papers you mentioned in appendix A.5 (Tables 2 & 3), taking inspiration from the work of Bu et al (2023). Additionally, we see that our approach is complementary to the other methods you mentioned, scaling differently with respect to hyper-parameters.
>
> |                           | Instantiating per-sample gradient | Storing every layer's gradient | Instantiating non-DP gradient | Number of back-propagations | Overhead independent of batch size |
> |---------------------------|-----------------------------------|---------------------------------|-------------------------------|-----------------------------|-----------------------------------|
> | non-DP                    | ❌                              | ❌                             | ✅            | 1                           | ✅                |
> | TF-Privacy | ✅ | ❌                       | ✅            | B                           | ❌                               |
> | Opacus     | ✅                 | ✅              | ✅            | 1                           | ❌                               |
> | FastGradClip | ✅           |            ❌                 | ❌                           | 2                           | ❌                               |
> | GhostClip | ❌                    | ❌                             | ✅            | 2                           | ❌                              |
> | Book-Keeping | ❌                       | ❌                             | ❌                          | 1                           | ❌                               |
> | Clipless w/ Lipschitz (ours) | ❌                         | ❌                             | ✅            | 1                           | ✅                |
> | Clipless w/ GNP (ours)      | ❌                               | ❌                             | ✅            | 1                           | ✅                |
>
> |                           | Time Complexity Overhead | Memory Overhead            |
> |---------------------------|--------------------------|---------------------------|
> | non-DP                    | 0                        | 0                         |
> | TF-Privacy | $\mathcal{O}(BTpd)$       | 0                         |
> | Opacus    | $\mathcal{O}(BTpd)$           | $\mathcal{O}(Bpd)$             |
> | FastGradClip | $\mathcal{O}(BTpd)$        | $\mathcal{O}(Bpd)$             |
> | GhostClip | $\mathcal{O}(BTpd+BT^2)$ | $\mathcal{O}(BT^2)$             |
> | Book-Keeping | $\mathcal{O}(BTpd)$     | $\mathcal{O}(B\min(pd, T^2))$   |
> | Clipless w/ Lipschitz (ours) | $\mathcal{O}(Upd)$           | 0                         |
> | Clipless w/ GNP (ours)      | $\mathcal{O}(Upd+Vpd\min(p,d))$ | 0                         |
>
> We assume weight matrices of shape $p\times d$, and a *physical* batch size $B$. For images, $T$ is height$\times$width. $U$ is the number of iterations in Power Iteration algorithm, and $V$ the number of iterations in Bjorck projection. Typically $U,V<15$ in practice. The overhead is taken relatively to non-DP training *without* clipping.

---

> > ### Author Response · Authors · 2023-11-16
> >
> > ### Bias of Clipless DP-SGD
> >
> > > not to fully discuss the bias of Clipless DP-SGD
> >
> > We concur that Clipless DP-SGD also exhibits bias; however, we emphasize that these biases manifest in distinct forms.
> >
> > **The bias of the optimizer:** in DP-SGD (with clipping) the average gradient is biased by the elementwise clipping. Therefore, the clipping may slow down convergence or lead to sub-optimal solutions.
> >
> > **The bias of the model in the space of Lipschitz networks:** this is a bias of the model, not of the optimizer. It has been shown that any classification task could be solved with a 1-Lipschitz classifier (Bethune et al, 2022), and in this sense, the bias induced by the space of 1-Lipschitz functions is not too severe. Better, this bias is precisely what allows us to produce robustness certificates (see for example Yang et al (2020) or our Figure 4).
> >
> > Yang, Y.Y., Rashtchian, C., Zhang, H., Salakhutdinov, R.R. and Chaudhuri, K., 2020. *A closer look at accuracy vs. robustness.* Advances in neural information processing systems, 33, pp.8588-8601.
> >
> > **Finally, there is the last source of bias:** the implicit bias induced by a given architecture on the optimizer, which can have strong effects on effective generalization. For neural networks (Lipschitz or not), this implicit bias is not fully understood yet. But even on large Lipschitz models, it seems that the Lipschitz constraint biases the network toward better robustness radii but worse clean accuracy, as frequently observed in the relevant literature.
> >
> > Therefore, these two biases influence the learning process differently, and they constitute two distinct (not necessarily exclusive) approaches. We added a discussion with figure 7 in appendix A.4.

---

> > > ### Comment · Reviewer_GMKM · 2023-11-22
> > >
> > > I will keep my score. The authors are mistaken in claiming "To the best of our knowledge, these new techniques are not (yet) included in these libraries." They each have their own library. GhostClip is published in 2022 at https://github.com/lxuechen/private-transformers. Book-Keeping is published in 2022 at https://github.com/awslabs/fast-differential-privacy. In addition, I also think the (additional) hyper-parameter tuning is a privacy concern and a difficulty in system design, which limit the applicability.

---

> > > > ### Author Response · Authors · 2023-11-23
> > > >
> > > > > "The authors are mistaken in claiming [..] these new techniques are not (yet) included in these libraries"
> > > >
> > > > Dear reviewer, we want to clarify our previous answer.
> > > > We want to reassure you that we are well aware that Book-Keeping and GhostClip have their libraries (the code is linked in the papers), and our answer is not a criticism addressed toward them.
> > > >
> > > > In our comment, "these" referred to "TF-privacy, Opacus, and jax-privacy" - not the ones you mention. Maybe our wording was a bit confusing.
> > > >
> > > > In the paper, Clipless DP-SGD is not presented as a strict improvement over GhostClip or Book-Keeping, but rather as an complementary solution, thanks to a unique property:
> > > > "where the **computational overhead is independent of the batch size** ".

---

### Official Review · Reviewer_L744 · 2023-11-01

**Soundness:** 3 good
**Presentation:** 4 excellent
**Contribution:** 3 good
**Rating:** 8
**Confidence:** 3

**Summary:**

In this work, the authors introduce Clipless DP-SGD, an approach for training neural network models with differentially private stochastic gradient descent, without the need to apply clipping to the parameter gradients.
To achieve this, the authors leverage existing theory on Lipschitz neural networks, which they extend to compute Lipschitz constants with respect to the parameters (instead of w.r.t. the inputs, which has already been studied in the literature).
This enables them to compute per-layer sensitivities, for a range of common feedforward architectures, without requiring gradient clipping.
The authors evaluate their approach on a range of benchmark datasets, where they observe a competitive performance compared to DP-SGD.

**Strengths:**

In my view the main strengths of the paper are as follows:

__Eliminating the need to tune $C$ and reducing gradient bias:__
One important benefit of Clipless DP-SGD is that it does not involve a clipping threshold $C,$ and as such it does not require tuning this parameter.
This is particularly useful since, to extract strong performance out of DP-SGD, selecting an appropriate value for $C,$ i.e. tuning it, is necessary.
Performing this tuning naively would result in privacy leakage since updating $C$ requires querying the data.
Clipless DP-SGD elegantly circumvents this issue by eliminating clipping altogether.
Another benefit of removing clipping, as the authors note, is that this eliminates the associated bias that is introduced in the gradients, potentially making optimisation easier, though to my understanding the effect of eliminating this bias has not been explicitly examined in the paper.

__Reduced runtime and memory requirements:__
Another benefit of the proposed method is the fact that, while DP-SGD computes per-sample gradients and clips these separately before averaging, Clipless DP-SGD operates on the averaged loss directly, without computing per-sample gradients.
As a result, Clipless DP-SGD offers both a lower memory footprint as well as faster runtimes than regular DP-SGD.
The favourable runtime and memory requirements of the proposed method are illustrated in figure 3 (though it should be noted that different implementations on different back-ends may not be directly comparable).

__Originality and relations to existing work:__
In my assessment the contribution made in this paper is both original and creative, since computing Lipchitz constants with respect to the network parameters does not seem to have been studied in the literature before.
The authors build on existing literature on Lipchitz networks and extend it in a valuable way.

__Codebase contribution:__
The authors provide a codebase for differentially private Lipchitz models, $\texttt{lip-dp}$ which is a valuable contribution in itself, and may be especially useful to practitioners.

__Quality of exposition and rigour:__
I found the main text well written and relatively easy to read considering its technical content.
The method is well motivated and well explained, and although I have not examined the proofs very closely, the exposition in the main text seems sound.

**Weaknesses:**

In my view, the paper does not present critical weaknesses, though two points that I think are important to consider are:

__Performance compared to existing methods:__
It seems that, while the current method eliminates the need for clipping (thereby simplifying the tuning procedure and removing gradient bias), it still performs worse than existing methods on datasets beyond MNIST, sometimes by a large margin (see figure 13 in the appendix).
It is not entirely clear which design choice is responsible for this gap in performance, though the Lipschitz constraints imposed on the network is a likely candidate.
Given this, it is not entirely clear how competitive the proposed method would be compared to existing methods, on more realistic tasks.

__Hyperparameter tuning:__
While Clipless DP-SGD removes the need for tuning the clipping threshold, it still appears to require a large amount of hyperparameter tuning in order to extract good performance (see figure 3 in the main text, and figure 13 in the appendix).
As a result, this can lead to increased computational costs as well as privacy leakage, since the current method does not account for leakage due to hyperparameter tuning.
It is not fully obvious how Clipless DP-SGD compares to existing methods under a like-for-like compute resources and privacy leakage due to tuning.

**Questions:**

I have the following questions for the authors regarding this work:

__Clarification on pareto fronts:__
To my understanding, the pareto fronts shown correspond to the convex hull of the green points.
The authors explain that the green points themselves correspond to a pair of validation accuracy and $\epsilon$ parameter from a given epoch.
Do these points correspond to all epochs across all Bayesian optimisation runs, as explained in appendix D2?
Can the authors comment on the extent of the privacy leakage that would result from selecting a particular model from the pareto front?

__Extending this to other architectures, e.g. transformers:__
The approach developed in this paper applies to feedforward networks, which admittedly cover a significant range of existing architectures.
Can the authors comment on whether it is possible to extend their method to other popular architectures, such as transformers or graph neural networks?
It is unclear to me how the ideas developed here can be extended to, for example, cases where the internal representation of the network depends on the input datum itself.

__How tight are the derived sensitivities in practice?__
Can the authors comment on how tight the derived sensitivities are in practice?
In particular, how large are the gradient norms encountered during training, compared to the derived sensitivities.


In addition, I would like to point out the following typos and suggestions:

- __Typo:__
In def. 3, "there is no parameters" should be "there are no parameters."
- __Suggestion:__
It might be worth separating the definition of Lipschitz networks from regular feedforward networks, keeping two separate definitions, or making the first part of the definition part of the text.
- __Typo & suggestion:__
In Theorem 1, "centered in zero" should be "centered at zero."
Also you could change "expanded analytically." to "computed analytically, as follows:"
- __Typo:__
In page 7, "for Lipschitz constrained network" should be "for Lipschitz constrained networks".

---

> ### Author Response · Authors · 2023-11-16
> **Thank you for your detailed review and your kind words**
>
> We applied your suggested modifications. We address two of your questions in the common rebuttal, and the remaining ones below.
>
> > “How tight are the derived sensitivities in practice?”
>
> We performed an empirical evaluation in Appendix **C.3** and Figures **11 & 12**. We measured the ratio between the empirical norm and the upper bound on the layer of a single-block MLP Mixer on Cifar-10. At the beginning of training, on the last layers, the ratio is between 30% and 40%. As we go closer to the first layers, the ratio falls to 10%. At the end of training, the ratio of this same layer is 4%. This is expected as the gradient norm tends to diminish during training, as we get closer to the (local) optimum.  We hypothesized it was due to the looseness of Cauchy-Schwartz inequality on $\frac{\partial f_d(x_d,\theta_d)}{\partial \theta}$ term, and the approximations during the "propage input bounds" step.
>
> > “it still appears to require a large amount of hyperparameter tuning in order to extract good performance”
>
> We want to reassure you that this is not the case: we reported all experiments for exhaustivity reasons, but the Bayesian optimizer reaches the Pareto front within 30 repetitions on image classification tasks. We observed that Clipless DP-SGD is not more sensitive than vanilla DP-SGD.

---

> > ### Comment · Reviewer_L744 · 2023-11-22
> > **Response to rebuttal**
> >
> > Thank you for your rebuttal.
> > I think this addresses the questions I raised sufficiently well.
> > I will maintain my current score.

---

### Official Review · Reviewer_ah9n · 2023-11-04

**Soundness:** 3 good
**Presentation:** 2 fair
**Contribution:** 3 good
**Rating:** 6
**Confidence:** 4

**Summary:**

This paper presents DP-SGD without gradient clipping, achieved through gradient norm preservation, a method that has not been explored in the existing DP-SGD literature.

**Strengths:**

This paper introduces a novel approach in the DP-SGD literature by offering DP-SGD without clipping through gradient norm preservation, a method that has not been explored before.

**Weaknesses:**

To validate the effectiveness of the proposed methods, it is essential to conduct a more comprehensive experimental evaluation.

**Questions:**

How can we ensure a fair experimental comparison? Specifically, tuning hyperparameters also consumes the DP budget. Did you take into account such DP costs in your experiments and results?

---

> ### Author Response · Authors · 2023-11-16
> **Thank you for your assessment of our work**
>
> Your question about the privacy cost of the hyper-parameter search is a valid concern we address in the common answer. As explained in Sec 4.1, we are aware of the risks associated to hyper-parameters tuning, and we ensured that the same number of repetitions was used for both algorithms when performing hyper-parameter search. We added this precision to the manuscript.

---

### Author Response · Authors · 2023-11-16
**Answers to frequent questions**

We thank all reviewers for their insightful comments. There is a consensus on the qualities of our work, praising both its **originality**:  “a method that has not been explored before” (R ah9n), “the contribution made in this paper is both original and creative“ (R L744); its **clarity**:  “Quality of exposition and rigor” (R L744) “the paper is well-polished and rigorous” (R GMKM); and its **potential**: “Empirical results on toy datasets show some promise for this new method” (R GMKM).

### On scaling to larger models

A common question was the scalability of the method to larger models.

> “Extending this to other architectures, e.g. transformers” (Reviewer L744)

**Transformer and self-attention:**  A recurring question concerned the Transformer architecture.

The "embedding layer” of transformer architectures (mapping discrete tokens to vectors) can straightforwardly benefit from the approach since it implements the function $\theta\mapsto \theta$ (1-Lipschitz). Unfortunately, vanilla transformers are not Lipschitz due to the product term $KQ$ appearing in self-attention (Kim et al, 2021). Previous works attempted to solve it, see for example Xu et al (2022). However, these constructions differ significantly from the original self-attention, and it is unclear if a large ViT backbone can be built with these.

Kim, H., Papamakarios, G. and Mnih, A., 2021, July. *The lipschitz constant of self-attention*. In International Conference on Machine Learning (pp. 5562-5571). PMLR.

Xu, X., Li, L., Cheng, Y., Mukherjee, S., Awadallah, A.H. and Li, B., 2022. *Certifiably Robust Transformers with 1-Lipschitz Self-Attention*. openreview preprint

Note that the MLP Mixer architecture we implemented is Lipschitz, and operates on patches like the ViTs do: they have a small number of parameters but require a huge amount of flops.

**Large scale pretraining**:
> “I wonder whether the new method can benefit from large-scale pretraining?” (Reviewer GMKM)

It can! In Appendix D.4 we did a preliminary experiment by testing a simple MLP on top of the latent space of a MobileNet. However, for fine-tuning, we need a large K-Lipschitz pre-trained backbone. To the best of our knowledge, this does not exist yet, but the recent work of Ghazanfari (2023) is promising.

Ghazanfari, S., Araujo, A., Krishnamurthy, P., Khorrami, F. and Garg, S., 2023. *LipSim: A Provably Robust Perceptual Similarity Metric*. arXiv preprint arXiv:2310.18274.

Training and fine-tuning a large Lipschitz vision model is beyond the scope of our work, which focuses on methodological contributions.


### On privacy leakages of hyper-parameter search

A common concern among reviewers was about the privacy cost of the hyper-parameter search and the sensitivity of the method to hyper-parameters in general.

> “ tuning hyperparameters also consumes the DP budget. “ (Reviewer ah9n)
> “Can the authors comment on the extent of the privacy leakage that would result from selecting a particular model from the pareto front?” (Reviewer L744)

Firstly, we emphasize that the vast majority of papers in the field ignore the privacy cost induced by the hyper-parameter search, and we adopted this common practice in reporting our results.

Secondly, some recent works suggest that it is possible to output one of the best models with a privacy cost logarithmic or even independent from the number of repetitions. We added these references in the final manuscript.

Papernot, N. and Steinke, T., 2021, October. *Hyperparameter Tuning with Renyi Differential Privacy*. In International Conference on Learning Representations.

Ding, Y. and Wu, X., 2022, November. *Revisiting Hyperparameter Tuning with Differential Privacy*. In NeurIPS ML Safety Workshop.

Finally, we would like to clarify that Clipless DP-SGD is *not* more sensitive than vanilla DP-SGD. We reported all the experiments across a broad range of hyperparameters for the sake of exhaustiveness. But the number of crucial hyper-parameters is not higher. We performed the same number of repetitions for the baseline DP-SGD and for Clipless DP-SGD.

### Paper revision

We incorporated the modifications and clarifications in the paper and its appendix, highlighted in **blue**. We warmly thank all the reviewers for their detailed feedback.

---

### Meta-Review · Area_Chair_MzDb · 2023-12-12

**Metareview:**

(a) Summarize the scientific claims and findings of the paper based on your own reading and characterizations from the reviewers.

The theoretical analysis reveals an unexplored link between the Lipschitz constant with respect to their input and the one with respect to their parameters. By bounding the Lipschitz constant of each layer with respect to its parameters, the authors train these networks with privacy guarantees. The analysis not only allows the computation of the aforementioned sensitivities at scale, but also provides guidance on how to maximize the gradient-to-noise ratio for fixed privacy guarantees.

(b) What are the strengths of the paper?

One important benefit of Clipless DP-SGD is that it does not involve a clipping threshold C and as such it does not require tuning this parameter. As a result, Clipless DP-SGD offers both a lower memory footprint as well as faster runtimes than regular DP-SGD.  The contribution made in this paper is both original and creative, since computing Lipchitz constants with respect to the network parameters is novel.

(c) What are the weaknesses of the paper? What might be missing in the submission?

The proposed method still performs worse than existing methods on datasets beyond MNIST, sometimes by a large margin. While Clipless DP-SGD removes the need for tuning the clipping threshold, it still appears to require a large amount of hyperparameter tuning in order to extract good performance.

**Justification For Why Not Higher Score:**

The empirical results in the paper fall short of convincing us that the proposed method is practically attractive. More empirical results with further gain will be needed.

**Justification For Why Not Lower Score:**

The paper has enough interesting ideas with novel solutions to merit publication. Especially the motivation for clipless DPSGD and the proposed solution are perfectly matched.

---

### Decision · Program_Chairs · 2024-01-16

Accept (poster)